# Inhibition of p38-MK2 pathway enhances the efficacy of microtubule inhibitors in breast cancer cells

Yu-Chia Chen[1,2†], Mamoru Takada[3†], Aerica Nagornyuk[4], Muhan Yu[1,3], Hideyuki Yamada[3], Takeshi Nagashima[3], Masayuki Ohtsuka[3], Jennifer G DeLuca[5], Steven M Markus[5], Motoki Takaku[4], Aussie Suzuki[1,2,6]*

[1]McArdle Laboratory for Cancer Research, Department of Oncology, University of Wisconsin-Madison, Madison, United States; [2]Molecular Cellular Pharmacology Graduate Program, University of Wisconsin-Madison, Madison, United States; [3]Department of General Surgery, Graduate School of Medicine, Chiba University, Chiba, Japan; [4]Department of Biomedical Science, University of North Dakota School of Medicine and Health Science, Grand Forks, United States; [5]Department of Biochemistry and Molecular Biology, Colorado State University, Fort Collins, United States; [6]Carbone Comprehensive Cancer Center, University of Wisconsin-Madison, Madison, United States

*For correspondence: aussie.suzuki@wisc.edu

†These authors contributed equally to this work

Competing interest: The authors declare that no competing interests exist.

## eLife Assessment

This study provides **valuable** findings that MK2 inhibitor CMPD1 can inhibit the growth, migration and invasion of breast cancer cells both in vitro and in vivo. The evidence supporting the claims of the authors is **solid**, although the detailed molecular mechanism and additional animal experiments would strengthen the paper. This study will be of interest to the breast cancer field.

**Abstract** Microtubule-targeting agents (MTAs) are widely used as first- and second-line chemotherapies for various cancers. However, current MTAs exhibit positive responses only in subsets of patients and are often accompanied by side effects due to their impact on normal cells. This underscores an urgent need to develop novel therapeutic strategies that enhance MTA efficacy while minimizing toxicity to normal tissues. Here, we demonstrate that inhibition of the p38 MAPK-MK2 signaling pathway sensitizes cancer cells to MTA treatment. We utilize CMPD1, a dual-target inhibitor, to concurrently suppress the p38-MK2 pathway and microtubule dynamicity. In addition to its established role as an MK2 inhibitor, we find that CMPD1 rapidly induces microtubule depolymerization, preferentially at the microtubule plus end, leading to the inhibition of tumor growth and cancer cell invasion in both *in vitro* and *in vivo* models. Notably, 10 nM CMPD1 is sufficient to induce irreversible mitotic defects in cancer cells, but not in non-transformed normal cells, highlighting its high specificity to cancer cells. We further validate that a specific p38-MK2 inhibitor significantly potentiates the efficacy of subclinical concentrations of MTA. In summary, our findings suggest that the p38-MK2 pathway presents a promising therapeutic target in combination with MTAs in cancer treatment.

## Introduction

Cancer, a disease characterized by uncontrolled cell growth, results from cells that proliferate indefinitely without external growth signals (*Hanahan and Weinberg, 2011*; *Aaronson, 1991*). Consequently, targeting cell cycle progression is a powerful therapeutic strategy for cancer treatment (*Chan et al., 2012*; *Manchado et al., 2012*). Microtubule-targeting agents (MTAs) disrupt spindle microtubule assembly during mitosis, making them widely used chemotherapy drugs for various tumors (*Matson and Stukenberg, 2011*; *Tischer and Gergely, 2019*; *Mukhtar et al., 2014*). By impairing microtubule dynamics and functions, MTAs activate the spindle assembly checkpoint (SAC) (*Musacchio, 2015*; *Lara-Gonzalez et al., 2021*; *Musacchio and Salmon, 2007*), leading to mitotic arrest, mitotic defects, and apoptotic cell death (*Matson and Stukenberg, 2011*; *Jordan, 2002*). MTAs are functionally categorized into two groups: microtubule stabilizers (e.g. taxanes) (*Weaver, 2014*; *Schiff et al., 1979*) and microtubule depolymerizers (e.g. eribulin) (*Perez, 2009*; *Jordan et al., 2005*). Paclitaxel (PTX), a taxane, has been a highly successful anticancer drug in clinical use for over 30 years (*Weaver, 2014Abu Samaan et al., 2019*). However, several limitations hinder its therapeutic efficacy. First, only a subset of breast and ovarian cancer patients exhibits a favorable response to PTX (*Fountzilas et al., 2009*; *Holmes et al., 1991*; *Kohn et al., 1994*). Second, PTX lacks tumor specificity, leading to off-target effects, including neutropenia, gastrointestinal disorders, and peripheral neuropathy (*Brewer et al., 2016*; *Armstrong et al., 2006*; *Banerji et al., 2014*; *Lipton et al., 1989*; *Misawa et al., 2023*). Third, cancer cells can acquire resistance to PTX through mechanisms such as the upregulation of drug efflux proteins (e.g. P-glycoprotein) or class III β-tubulin, which reduces PTX binding affinity (*Kavallaris et al., 1997*; *Kavallaris, 2010*; *Sève and Dumontet, 2008*). In contrast, eribulin, a second-line chemotherapeutic agent, has a lower incidence of peripheral neuropathy, one of the most troublesome side effects of MTAs (*Eslamian et al., 2017*). Due to its distinct mechanism of action, eribulin is effective against taxane-resistant tumors (*Smith et al., 2010*; *O'Shaughnessy et al., 2015*; *Dumontet and Jordan, 2010*). However, clinical studies have demonstrated that fewer than 20% of metastatic breast cancer patients, previously treated with chemotherapy, respond positively to eribulin (*Cortes et al., 2011*; *Cortes et al., 2010*). MTAs remain central to the treatment of breast and ovarian cancers, including metastatic cases. Therefore, developing new strategies to overcome the limitations associated with MTAs is critical for optimizing therapeutic outcomes in cancer treatment.

The p38 mitogen-activated protein kinase (MAPK) signaling pathway, activated by a variety of environmental and intracellular stimuli (*Wood et al., 2009*; *Borisova et al., 2018*; *Yamashita et al., 2008*; *Han et al., 1994*), plays a crucial role in numerous biological processes, including DNA repair, inflammation, cell differentiation, and cell death (*Colomer et al., 2019*; *Arthur and Ley, 2013*; *Segalés et al., 2016*; *Canovas and Nebreda, 2021*; *Cuadrado and Nebreda, 2010*; *Young, 2013*). MAPK-activated protein kinase 2 (MK2) is a major downstream substrate of p38 MAPK (*Gurgis et al., 2014*). Previous studies have shown that phosphorylated MK2 localizes to mitotic spindles, and that MK2 depletion leads to abnormal spindle formation, defects in chromosome alignment, and mitotic arrest in both human cells and mouse oocytes, indicating a vital role of MK2 in mitotic progression (*Yuan et al., 2010*; *Tang et al., 2008*). CMPD1 was originally developed as a selective inhibitor targeting the p38-dependent phosphorylation of MK2 (*Figure 1A*; *Davidson et al., 2004*; *Mamidi et al., 2021*; *Wu et al., 2018*; *Hendriks et al., 2010*). Subsequent studies revealed that CMPD1 could induce G2/M arrest in glioblastoma and gastric cancer cells, as observed by flow cytometry (*Li et al., 2018*; *Gurgis et al., 2015*), and suggested its potential role in inhibiting microtubule polymerization *in vitro* (*Gurgis et al., 2015*). However, the precise and dynamic nature of CMPD1's effects on microtubule dynamics and cancer cell proliferation has yet to be fully elucidated. In this study, we demonstrate that CMPD1 preferentially induces severe mitotic defects in breast cancer cells and effectively inhibits cancer cell growth, migration, and invasion in both *in vitro* and *in vivo* models. Notably, CMPD1 uniquely triggers rapid microtubule depolymerization at the plus end *in vitro*. These results indicate that the inhibition of the p38-MK2 pathway could enhance the therapeutic efficacy of MTAs. We validate this hypothesis by demonstrating that a specific MK2 inhibitor, MK2-IN-3, in combination with vinblastine, a clinically approved microtubule destabilizer (*Jordan and Wilson, 2004*; *Moudi et al., 2013*), exhibits significantly increased efficacy in inducing mitotic defects. Thus, our results suggest that the p38-MK2 pathway may serve as a promising therapeutic target in combination with MTAs in cancer treatment.

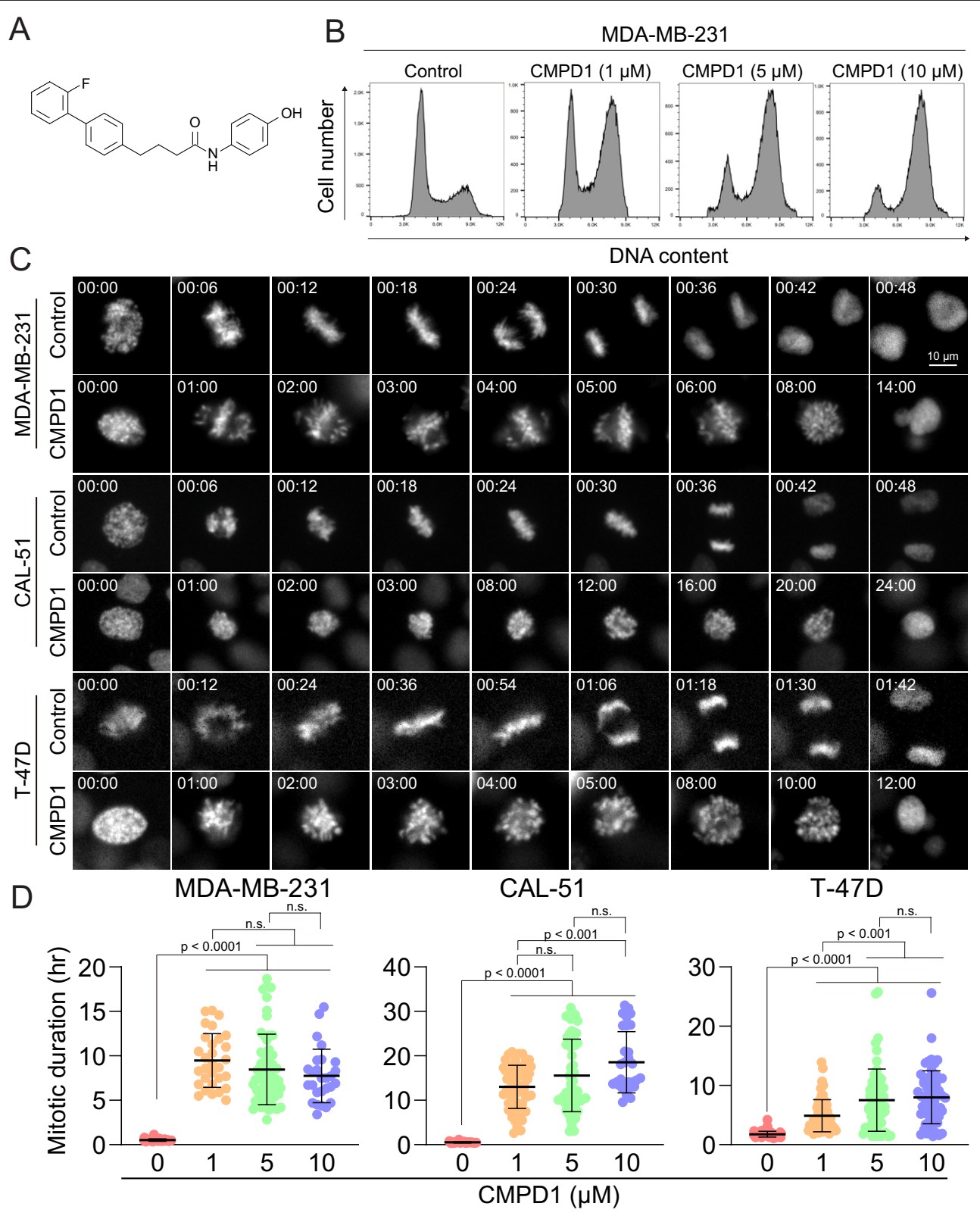

**Figure 1.** CMPD1 induced severe mitotic arrest in multiple cancer cell lines. (**A**) Chemical structure of CMPD1. (**B**) FACS analysis of MDA-MB-231 cells treated with DMSO or 1, 5, or 10 μM CMPD1 for 24 hr. (**C**) Representative time-lapse images of MDA-MB-231, CAL-51, and T-47D cells treated with either DMSO or 5 μM CMPD1. Time is indicated in minutes post-nuclear envelope breakdown (NEBD). (**D**) Quantification of mitotic duration of MDA-MB-231, CAL-51, and T-47D treated with DMSO, 1, 5, or 10 μM CMPD1. n=30–60 cells pooled from two biological replicates. Results are the mean ± standard deviation (s.d.). The p-value was calculated using Tukey's multiple comparison test.

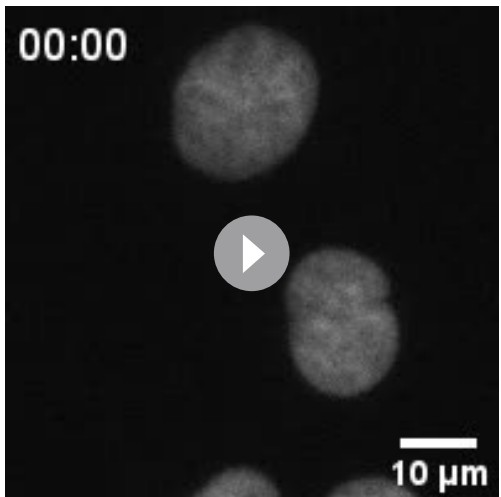

**Video 1.** Control MDA-MB-231 (H2B-mCherry) mitotic cells.
https://elifesciences.org/articles/104859/figures#video1

## Results

### CMPD1 induces robust prometaphase arrest in breast cancer cell lines

Cell cycle inhibitors, particularly MTAs, are standard chemotherapy drugs for breast cancer (*Jordan and Wilson, 2004*). To investigate whether CMPD1 induces G2/M arrest in breast cancer cells, we treated MDA-MB-231 cells, a triple-negative breast cancer (TNBC) cell line, with varying concentrations of CMPD1 and analyzed its effects on cell cycle progression using flow cytometry. The results demonstrated that CMPD1 effectively arrested MDA-MB-231 cells in the G2/M phase at concentrations ranging from 1 to 10 µM (*Figure 1B*), consistent with previous observations in glioblastoma cells (*Gurgis et al., 2015*). Given that flow cytometry is unable to distinguish between G2 phase and mitosis, we performed high-temporal resolution live-cell imaging to further dissect the impact of CMPD1 on cell cycle progression across various breast cancer cell lines, including MDA-MB-231 (TNBC), CAL-51 (TNBC), and T-47D (luminal A) (*Dai et al., 2017*; *Neve et al., 2006*). Our findings revealed that CMPD1 induced a severe prometaphase arrest across all tested breast cancer cell lines, regardless of subtypes, with most cells remaining arrested in prometaphase for over 10 hr, while control cells divided within 30–60 min (*Figure 1C and D* and *Videos 1–2*). Since CAL-51 cells harbor wild-type TP53, whereas the other two cell lines do not (*Huovinen et al., 2011*; *Tentler et al., 2015*), these results suggest that CMPD1 induces a robust prometaphase arrest in breast cancer cells through a p53-independent mechanism.

### Breast cancer cells exhibit heightened sensitivity to CMPD1 treatment compared to normal cells

We demonstrated that concentrations exceeding 1 µM of CMPD1 effectively induced a robust prometaphase arrest in multiple breast cancer cell lines (*Figure 1D*). In contrast, recent studies on MTA treatment for breast cancer have identified that the clinically relevant concentration of PTX in tissue culture cells ranges from 5 to 50 nM (*Scribano et al., 2021*). Within this range of concentration, PTX does not induce severe mitotic arrest as observed at higher concentrations (>1 µM), but it significantly increases the incidence of mitotic errors, thereby promoting chromosomal instability (CIN) and ultimately leading to cancer cell death (*Scribano et al., 2021*). Given these findings, we next explored whether sub-µM concentrations of CMPD1 could induce CIN in breast cancer cells. To this end, we treated MDA-MB-231 and CAL-51 cells, along with two non-transformed cell lines (breast epithelial MCF10A cells and retinal pigment epithelial RPE1 cells), which served as a normal cell control, with either 10 or 50 nM CMPD1. We then compared the impact of CMPD1 on chromosome segregation using live-cell imaging. In the absence of CMPD1, no significant differences in mitotic duration were observed among MCF10A, RPE1, MDA-MB-231,

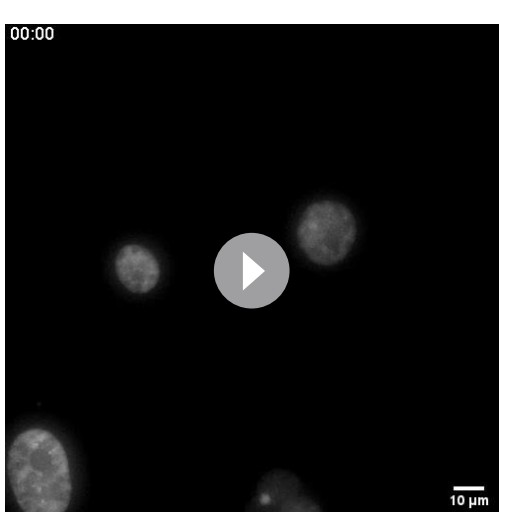

**Video 2.** CMPD1-treated MDA-MB-231 (H2B-mCherry) mitotic cells.
https://elifesciences.org/articles/104859/figures#video2

and CAL-51 cells (*Figure 2A–C* and *Figure 2—figure supplement 1A–C*). However, upon treatment with 10 or 50 nM CMPD1, all cell lines were arrested in prometaphase, except for MCF10A (*Figure 2A–C*). Notably, MDA-MB-231 and CAL-51 cells displayed significantly prolonged mitotic duration compared to MCF10A and RPE1 cells (*Figure 2A–C* and *Figure 2—figure supplement 1A–C*), indicating greater sensitivity of breast cancer cells to CMPD1 treatment relative to normal cells. Given that mitotic errors are a direct cause of CIN, we next assessed the frequency of mitotic errors, including misaligned chromosomes, chromosome bridges, lagging chromosomes, and multipolar division under these conditions (*Figure 2D*). Both MDA-MB-231 and CAL-51 cells exhibited a significantly higher rate of mitotic errors when exposed to 10 or 50 nM CMPD1 (*Figure 2E*). For instance, approximately 90% and 100% of CAL-51 cells experienced mitotic errors in the presence of 10 and 50 nM CMPD1, respectively (*Figure 2E*). Conversely, 90–100% of MCF10A and RPE1 cells underwent faithful cell division even at the same concentrations of CMPD1 (*Figure 2E* and *Figure 2—figure supplement 1A–C*). To determine whether CMPD1 selectively targets cancer cells compared to other MTAs, we evaluated mitotic error rates in the same cell lines treated with a clinically relevant concentration of PTX (10 nM). Our results demonstrated that both RPE1 and breast cancer cell lines displayed comparably high rates of mitotic errors when treated with 10 nM PTX treatment (*Figure 2F*), consistent with previous observations (*Scribano et al., 2021*). In summary, unlike PTX, CMPD1 induces CIN with selective toxicity toward breast cancer cells.

To further investigate the selectivity of CMPD1 in breast cancer cells, we conducted a CMPD1 washout assay using the above cell lines, aiming to recapitulate the clinical condition where the concentration of chemotherapy drugs in patients is diluted due to the 'drug holiday' between regular treatments. In this assay, we treated the cells with 2 µM CMPD1 for 12 hr, followed by a washout before imaging. We observed that the mitotic duration was significantly prolonged in both MDA-MB-231 and CAL-51 cells upon CMPD1 washout (85 and 151 min, respectively) compared to RPE1 cells (14 min) (*Figure 2G and H* and *Figure 2—figure supplement 2A and B*). This suggests that breast cancer cells struggle to recover from CMPD1-induced mitotic arrest, whereas normal cells can immediately resume progression to proper anaphase. Notably, 90% and 81% of mitotic cells in MDA-MB-231 and CAL-51 cell lines, respectively, exhibited mitotic errors, whereas 50% of RPE1 cells displayed normal chromosome segregation (*Figure 2G and I*). These findings reinforce the hypothesis that breast cancer cells are more sensitive to CMPD1 treatment under clinically relevant conditions. Interestingly, approximately 60% of MDA-MB-231 cells that entered mitosis after CMPD1 washout still exhibited mitotic errors, while only ~10% of RPE1 cells showed errors. This suggests a prolonged impact of CMPD1 on mitosis in breast cancer cells (*Figure 2—figure supplement 2C*). Consistent with the increased sensitivity in breast cancer cells to CMPD1, MDA-MB-231 cells exhibited a significantly higher frequency of apoptotic cell death during or shortly after mitosis within 24 hr of CMPD1 washout, compared to RPE1 cells (*Figure 2—figure supplement 2D and E*).

To further validate the heightened sensitivity of breast cancer cells to CMPD1, we measured its $IC_{50}$ in MDA-MB-231, CAL-51, and MCF10A cells. Consistent with our live-cell imaging results, both MDA-MB-231 and CAL-51 cells exhibited significantly lower $IC_{50}$ (0.21 and 0.74 µM, respectively), compared to MCF10A cells (1.16 µM) (*Figure 2—figure supplement 3A–C*). Moreover, $IC_{50}$ of p53 knockout CAL-51 cells (*Redman-Rivera et al., 2021*) (0.88 µM) was comparable to that of wild-type CAL-51 cells (0.74 µM) (*Figure 2—figure supplement 3B and D*), further supporting the notion that CMPD1 induces a robust prometaphase arrest in breast cancer cells via a p53-independent mechanism. Collectively, these results underscore that breast cancer cells are more sensitive to CMPD1 treatment than normal MCF10A and RPE1 cells, particularly in a clinically relevant context.

## CMPD1 inhibits anchorage-independent growth and tumor growth in mouse xenograft

We subsequently evaluated the efficacy of CMPD1 in inhibiting anchorage-independent growth, a critical hallmark of tumorigenesis, in MDA-MB-231 cells. CMPD1 demonstrated a potent dose-dependent inhibition, with 500 nM being sufficient to completely suppress colony formation (*Figure 3A and B* and *Figure 3—figure supplement 1A and B*). Remarkably, even at a relatively lower concentration (250 nM), CMPD1 significantly reduced colony formation, surpassing the inhibitory effects of 10 µM PTX. These findings suggest that CMPD1 is more effective than PTX in inhibiting the anchorage-independent growth of MDA-MB-231 cells.

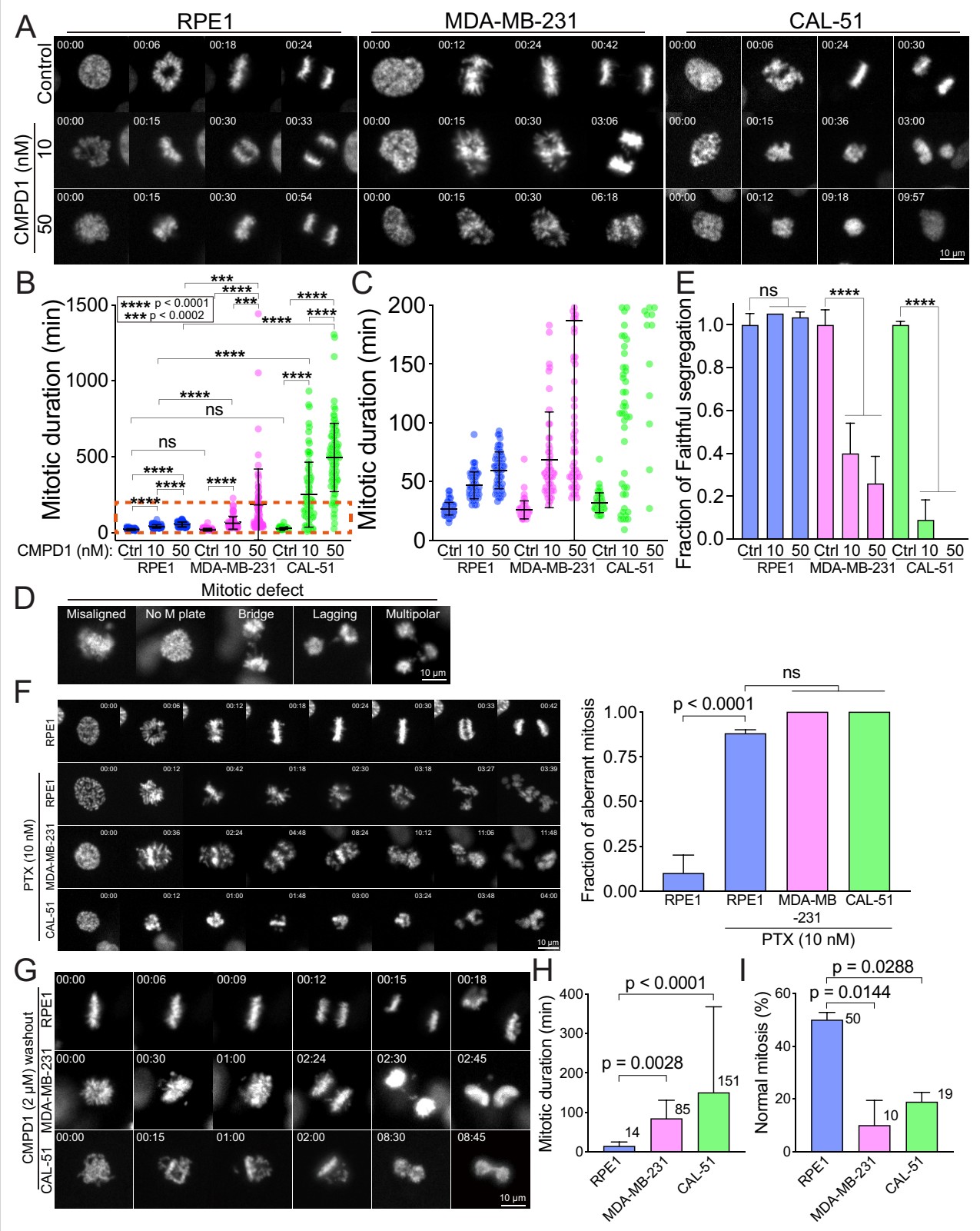

**Figure 2.** CMPD1 treatment specifically attenuates mitotic fidelity of cancer cells. (**A**) Representative time-lapse images (interval: 3 min) of RPE1, MDA-MB-231, and CAL-51 cells treated with DMSO or low dose of CMPD1 (10 and 50 nM). Time is indicated in minutes post-nuclear envelope breakdown (NEBD). (**B**) Quantification of mitotic duration of cells as shown in (**A**). n=70, 60, 60, 60, 60, 65, 50, 84, 101 cells from left to right. (**C**) The enlarged plot of a red box region as shown in (**B**). (**D**) Representative images of mitotic defects. Note that 'misaligned' and 'no metaphase (**M**) plate' were annotated

*Figure 2 continued on next page*

*Figure 2 continued*

only when the cells exhibited this phenotype upon anaphase onset. (**E**) Normalized fraction of mitotic cells exhibiting accurate chromosome segregation in each condition. The data derived from CMPD1-treated cells were normalized to the data derived from DMSO-treated cells for each cell line. (**F**) Left: representative time-lapse images (interval: 3 min) of untreated RPE1 cells, and RPE1, MDA-MB-231, and CAL-51 cells treated with 10 nM PTX. Right: the quantification of the fraction of mitotic cells showing mitotic errors. The p-value was calculated using Tukey's multiple comparisons test. Time is indicated in minutes post-washout. (**G**) Representative images of RPE1, MDA-MB-231, and CAL-51 cells in CMPD1 washout experiments. Briefly, cells were treated with 2 µM CMPD1 for 4 hr, followed by a wash with complete media and live-cell imaging. Time is indicated in minutes post-washout. (**H**) The mitotic duration of mitotic cells that were arrested by CMPD1 in the beginning of the imaging. The mitotic duration was defined as the time from the start of the imaging to anaphase onset or mitotic exit. The mean value was shown at the top right of each condition. n=54, 60, and 37 cells for RPE1, MDA-MB-231, and CAL-51 cells, respectively. The p-value was calculated using Tukey's multiple comparisons test. (**I**) Quantification of the percentage of mitotic cells showing normal chromosome segregation. The mean value was shown at the top right of each condition. n=50 cells in each condition pooled from two biological replicates. The p-value was calculated using Tukey's multiple comparisons test. The results are the mean ± s.d.

The online version of this article includes the following figure supplement(s) for figure 2:

**Figure supplement 1.** Low-dose CMPD1 does not decrease mitotic fidelity of MCF10A cells.

**Figure supplement 2.** Cancer cells exhibited higher sensitivity to CMPD1.

**Figure supplement 3.** Determination of CMPD1 $IC_{50}$ in MDA-MB-231, CAL-51, MCF10A, and p53 KO CAL-51 cell lines.

We demonstrated that CMPD1 effectively inhibits cancer cell growth in a tissue culture model. Next, we investigated whether CMPD1 could also inhibit tumor growth *in vivo*. To this end, we employed MDA-MB-231 and CAL-51 xenograft mouse models and found that CMPD1 significantly suppressed the tumor growth (*Figure 3C–G*, *Figure 3—figure supplement 1C and D*). Notably, while both PTX and CMPD1 suppressed tumor growth in mice, CMPD1 achieved comparable efficacy at concentrations 10–100 times lower than PTX (*Figure 3D–G*). Furthermore, mice in the CMPD1 treatment group continued to gain weight during the entire treatment regimen, comparable to the control group (vehicle-treated mice), with no observed mortality (*Figure 3—figure supplement 2A and B*). In contrast, the PTX-treated group exhibited a significant reduction in body weight and an increased number of deaths of individuals (*Figure 3—figure supplement 2A and B*). Additionally, anatomical examination showed no apparent abnormalities in kidneys or livers of CMPD1-treated mice, and blood marker analysis confirmed no significant impairment in these organ functions in CMPD1-treated mice compared to control (*Figure 3—figure supplement 3A-D*). Notably, the numbers of white blood cells (WBCs) were comparable between CMPD1-treated and control groups, while PTX-treated group showed a marked decrease in WBC levels (*Figure 3—figure supplement 3C and D*). These findings suggest that CMPD1 is not only effective in inhibiting tumor growth in both *in vitro* and *in vivo* models but also appears to be more potent and safer than PTX, a standard chemotherapeutic agent for breast cancer treatment.

## CMPD1 exhibits a preferential depolymerizing effect on the microtubule plus end

We demonstrated that at least ≥1 µM CMPD1 induced a robust prometaphase arrest in breast cancer cell lines (*Figure 1D*). A previous study suggests that CMPD1 may possess the ability to inhibit tubulin polymerization, as indicated by *in vitro* tubulin polymerization assay (*Gurgis et al., 2015*). To elucidate the mechanism underlying the CMPD1-induced prometaphase arrest, we performed high spatiotemporal resolution live-cell imaging to monitor microtubule dynamics in metaphase CAL-51 cells (see Materials and methods). In these cells, α-tubulin and histone H2B genes were endogenously labeled with mNeonGreen and mScarlet, respectively, using CRISPR-Cas9 technology (*Scribano et al., 2021*). Our results revealed that the signal intensity of mitotic spindles dramatically decreased to less than 10% within 8 min following CMPD1 treatment (*Figure 4A and B* and *Videos 3–4*). Concurrently, the formation of multipolar spindle poles was observed (*Figure 4A*). The decrease in spindle signal intensity correlated with a gradual disruption of the metaphase plate, likely due to the loss of kinetochore-microtubule attachments (*Figure 4A*). Moreover, we observed that CMPD1-treated cells exited the G2 phase and underwent nuclear envelope breakdown (NEBD), even though these cells were unable to assemble mitotic spindles (*Figure 4C*). This finding further supports that CMPD1 specifically arrests cells in prometaphase rather than G2 phase. To determine whether CMPD1 induces rapid microtubule depolymerization or the degradation of tubulin proteins, we pre-treated the cells with MG132, a proteasome inhibitor, for 1 hr prior to CMPD1 treatment (*Figure 4A and B* and *Video 5*). We

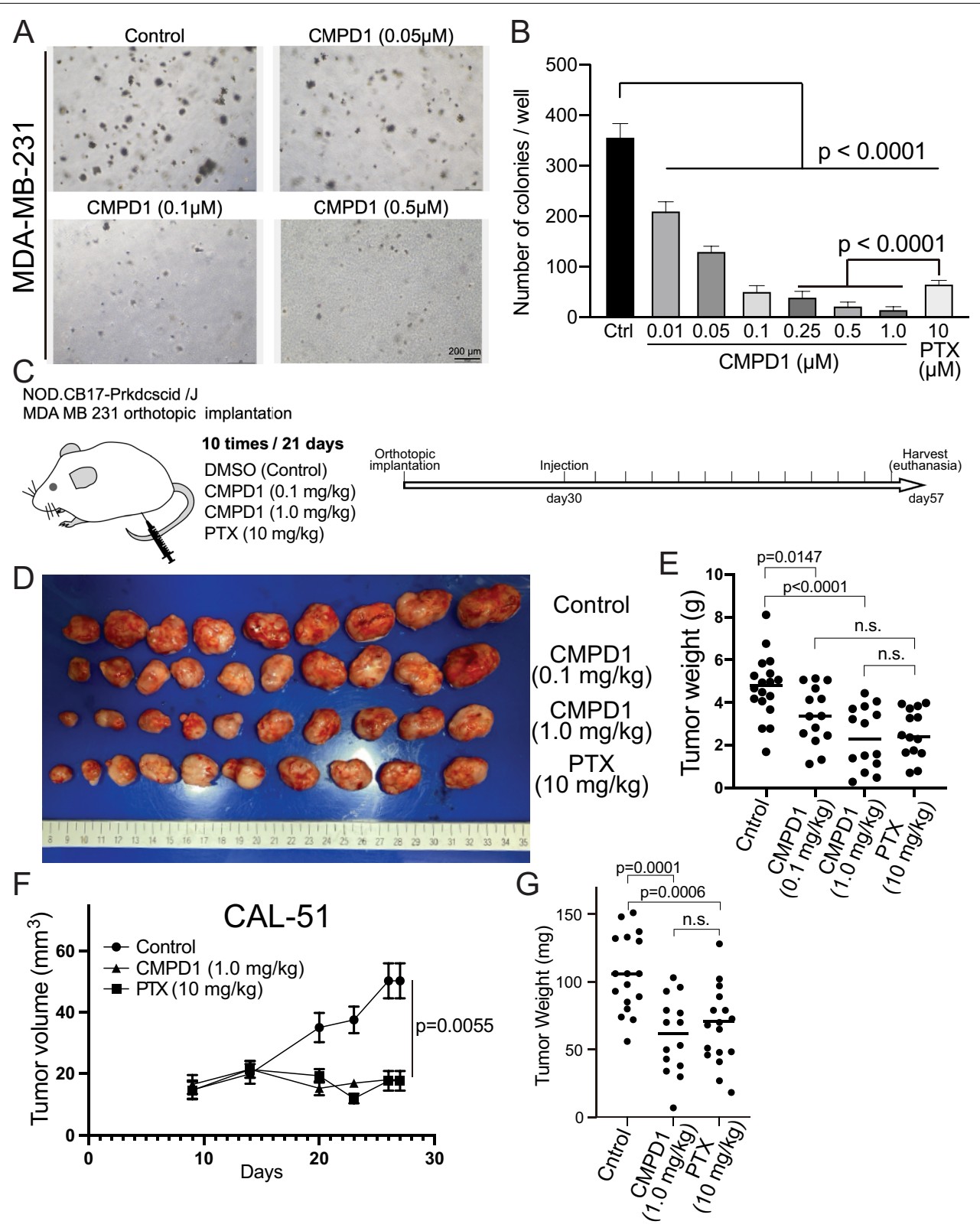

**Figure 3.** CMPD1 inhibits both anchorage-independent growth and tumor growth in mice. (**A**) Representative images of anchorage-independent growth assay using MDA-MB-231 cells treated with DMSO, CMPD1 at different concentrations (0.01, 0.05, 0.1, 0.25, 0.5, and 1 µM), or 10 µM paclitaxel (PTX). (**B**) The normalized number of cell colonies formed in each condition. Results are the mean ± s.d. Three biological replicates were performed for each condition. The p-value was calculated using Tukey's multiple comparison test. (**C**) Schematic diagram of the mouse xenograft experiment

*Figure 3 continued on next page*

*Figure 3 continued*

using MDA-MB-231 and the drug treatment schedule. (**D**) The photo of tumors in each condition at the time of necropsy. The tumors were arranged in order based on their size. (**E**) Quantification of the gross weight of tumors in the mouse xenograft experiment using MDA-MB-231 cells at the time of necropsy. The p-value was calculated using Tukey's multiple comparison test. (**F**) Quantification of the tumor volume during the treatment regimen in the mouse xenograft experiment using CAL-51 cells. (**G**) Quantification of the gross weight of tumors in the mouse xenograft experiment using CAL-51 cells at the time of necropsy. The p-value was calculated using Tukey's multiple comparison test.

The online version of this article includes the following figure supplement(s) for figure 3:

**Figure supplement 1.** CMPD1 inhibits tumor growth *in vitro* and *in vivo*.

**Figure supplement 2.** Body weight change and overall survival rate in mouse xenograft experiments.

**Figure supplement 3.** Histological and blood analyses in mouse xenograft experiments.

found that proteasome inhibition did not prevent the CMPD1-mediated reduction in spindle signals (*Figure 4A and B*), suggesting that CMPD1 directly induces microtubule depolymerization in breast cancer cells.

To further investigate the mechanism underlying CMPD1-induced microtubule depolymerization, we conducted total internal reflection fluorescence microscopy (TIRFM) experiments to observe microtubule behaviors in response to CMPD1 treatment. Although the conventional TIRFM assay using PTX-stabilized, immobilized microtubules is widely employed for its simplicity, it fails to recapitulate the dynamic behaviors of microtubules in living cells. This limitation hinders the assessment of CMPD1's effects under more physiologically relevant conditions. To overcome this limitation, we utilized GMPCPP-stabilized microtubule seeds combined with rhodamine-labeled tubulin, which allows for the observation of microtubule growth and shrinkage at both ends, thereby better mimicking the dynamic behavior of microtubules in mitotic cells. As expected, microtubules exhibited repeated cycles of growth and shrinkage at both plus and minus ends, with a higher stability at minus ends (*Figure 4D*). Upon treatment with CMPD1, there was a marked reduction in the proportion of microtubules with plus-end extensions within just 1 min (*Figure 4D–F*), along with an increased frequency of microtubule catastrophes, reduced growth rates, and decreased maximum lengths of plus-end extensions (*Figure 4G and H*). Interestingly, CMPD1's effects on the minus ends of microtubules were notably different. Neither the fraction of microtubules with minus-end extensions nor the catastrophe frequency at minus ends was significantly different from the control (*Figure 4D–H*). However, minor reductions were observed in the average maximum length of minus-end extensions and their growth rates (*Figure 4H*). In contrast, vinblastine, a well-characterized microtubule destabilizer, rapidly depolymerized microtubules from both ends (*Figure 4D and E*), highlighting that CMPD1 uniquely and preferentially depolymerizes microtubules from the plus ends. Collectively, our cell biology and biochemical data demonstrate that CMPD1 alone can depolymerize microtubules, with a pronounced preference for plus-end depolymerization.

## CMPD1 inhibits cell migration and invasion

Since proper microtubule dynamics is essential for regulating cell locomotion (*Vasiliev et al., 1970*; *Liao et al., 1995*; *Etienne-Manneville, 2013*), we hypothesized that the improper microtubule dynamics induced by CMPD1 might inhibit cell migration. To investigate this, we evaluated the migratory capacity of CAL-51 cells using a wound healing assay following treatment with CMPD1. Our findings revealed that CMPD1, at concentrations ranging from 100 nM to 10 µM, significantly inhibited wound closure compared to the DMSO-treated control (*Figure 5A and B*), demonstrating its potential to suppress cancer cell migration and invasion. To further examine this hypothesis, we conducted a transwell-invasion assay using MDA-MB-231 cells. Treatment with CMPD1 at concentrations greater than 100 nM significantly suppressed breast cancer cell invasion, and at 1 µM, CMPD1 completely abolished this cancer invasion (*Figure 5C and D*).

To determine whether CMPD1 inhibits cancer cell invasion *in vivo*, we assessed the frequency of invasion of cancer cells into blood vessels by examining the tissue sections of mice with cancer cell-derived xenografts described in *Figure 3C*. CMPD1-treated mice displayed a significantly reduced frequency of cancer cell-infiltrated vessels compared to vehicle-treated mice, suggesting that CMPD1 significantly inhibits metastasis *in vivo* (*Figure 5E*). Together, these results show that CMPD1 suppresses cancer cell migration and invasion *in vitro* and *in vivo*.

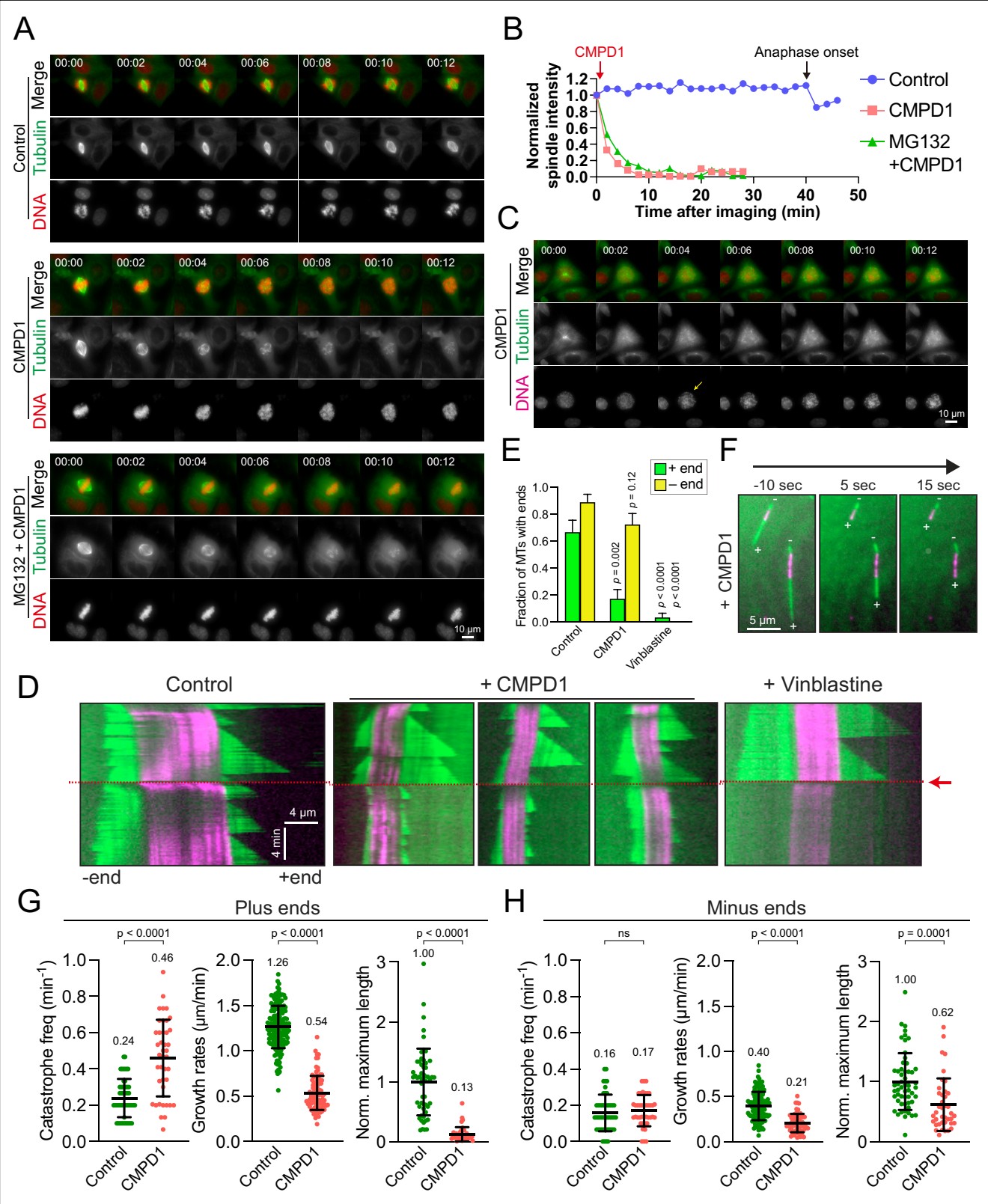

**Figure 4.** CMPD1 induces microtubule depolymerization. (**A**) Representative time-lapse images of CAL-51 cells expressing α-tubulin-mNeonGreen and H2B-mScarlet upon the treatment with DMSO, 2 μM CMPD1, or 10 μM MG132 along with 2 μM CMPD1. CMPD1 was added into the cell culture media immediately after images at the first time point were acquired. Time is indicated in minutes. (**B**) The quantification of the signal levels of mitotic spindles over time in each condition as shown in (**A**). (**C**) Representative time-lapse images (interval: 2 min) of a G2 phase CAL-51 cell expressing α-tubulin-

*Figure 4 continued on next page*

*Figure 4 continued*

mNeonGreen and H2B-mScarlet in the presence of 2 μM CMPD1. Note that the two bright dots in the tubulin channel indicate the two clustered centrosomes before nuclear envelope breakdown (NEBD). The arrow in the DNA channel indicates the time of NEBD. Time is indicated in minutes. (**D**) Representative kymographs depicting microtubule plus- and minus-end dynamics before and after the addition of polymerization mix supplemented with 15 μM tubulin alone, or 15 μM tubulin supplemented with either 20 μM CMPD1 or 5 μM vinblastine, as indicated (GMPCPP-stabilized microtubule seeds, magenta; microtubules polymerized from seeds, green; see Materials and methods). Plus ends are positioned to the right, and minus ends are positioned to the left of the seeds in all kymographs. Red arrow and dashed line indicate the time of addition of tubulin alone or tubulin plus drug. (**E**) Plot depicting the fraction of microtubules with detectable plus or minus ends 1 min after addition of drug, or tubulin alone (n=27, 29, and 30 microtubules from left to right). Results are the mean ± s.d. (**F**) Representative images from a time-lapse sequence showing microtubule plus- and minus-end lengths 10 s prior to, and 5 and 15 s after the addition of polymerization mix supplemented with 20 μM CMPD1. (**G**) Plots depicting plus-end catastrophe frequencies (n=51 and 41 microtubules from left to right), plus-end growth rates (n=172 and 91 events from left to right), and normalized maximum length (n=55 and 41 microtubules from left to right) achieved over the imaging period. (**H**) Plots depicting minus-end catastrophe frequencies (n=51 and 41 microtubules from left to right), minus-end growth rates (n=129 and 63 events from left to right), and normalized maximum length (n=55 and 40 microtubules from left to right) achieved over the imaging period. Results are the mean ± s.d. Data were pooled from at least two to four biological replicates (**G–H**).

## Inhibiting the p38-MK2 pathway significantly enhances the efficacy of microtubule depolymerizing agents

CMPD1's distinct capacity to inhibit tumor growth may stem from its dual inhibitory effects on both the p38-MK2 signaling pathway and microtubule dynamics. To rigorously test the hypothesis that inhibition of the p38-MK2 pathway could potentiate the efficacy of MTAs, we assessed the combinatorial effects of MK2-IN-3 (hereafter referred to as MK2i), a selective MK2 inhibitor (*Anderson et al., 2007*), and vinblastine at clinically relevant concentrations (1 or 5 nM) on mitotic progression in CAL-51 cells. We first confirmed that MK2i effectively inhibits MK2 activity in CAL-51 cells, as both 1 and 10 μM concentrations significantly reduced phosphorylation levels of Hsp27, a key downstream phosphorylation substrate of MK2 (*Stokoe et al., 1992*; *Lopes et al., 2009*), following $H_2O_2$-induced oxidative stress (*Figure 6—figure supplement 1A and B*). In contrast, treatment with 10 μM MK2i alone had no detectable impact on cell proliferation of CAL-51 cells over a 4-day period (*Figure 6—figure supplement 1C*). Consistent with these findings, treatment with either 10 μM MK2i or 1 nM vinblastine alone did not significantly alter the frequency of mitotic slippage or mitotic cell death compared to the control (*Figure 6A*). However, both individual treatments modestly prolonged mitotic duration and increased the frequency of mitotic errors (approximately two- to threefold compared to control) (*Figure 6A*). Strikingly, the combination of 10 μM MK2i with 1 nM vinblastine resulted in a profound synergistic effect, extending mitotic duration by approximately 13-fold and markedly increasing the frequency of mitotic errors and cell death (*Figure 6A*, Condition 4). A similar synergistic effect was

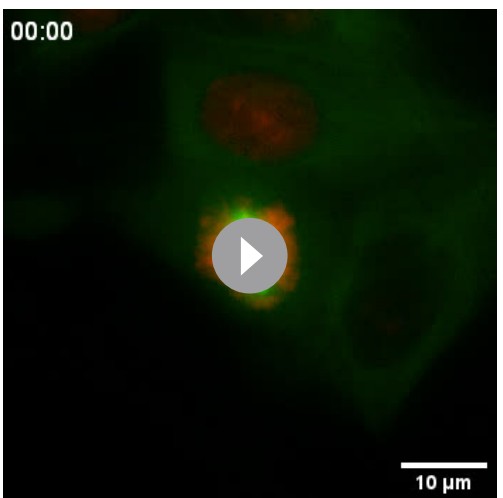

**Video 3.** Control CAL-51 (H2B-mScarlet and tubulin-mNeonGreen) mitotic cells.

https://elifesciences.org/articles/104859/figures#video3

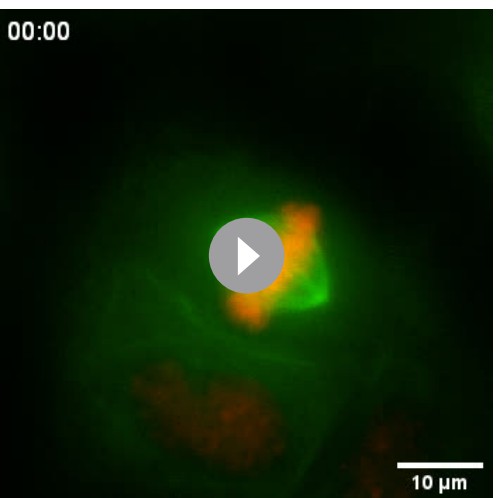

**Video 4.** CMPD1-treated CAL-51 (H2B-mScarlet and tubulin-mNeonGreen) mitotic cells.

https://elifesciences.org/articles/104859/figures#video4

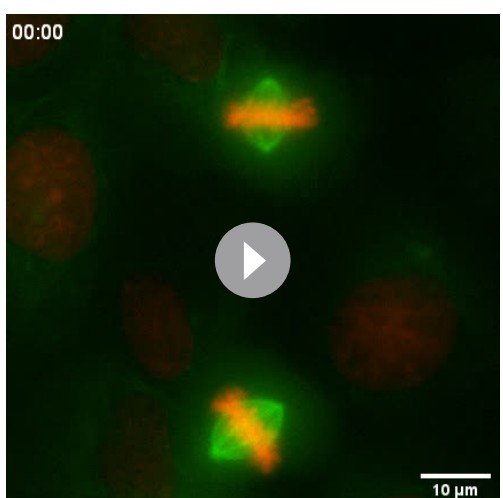

**Video 5.** MG132 and CMPD1-treated CAL-51 (H2B-mScarlet and tubulin-mNeonGreen) mitotic cells.
https://elifesciences.org/articles/104859/figures#video5

also observed when cells were simultaneously treated with 10 μM MK2i and 5 nM vinblastine, although the 5 nM vinblastine alone induced a more pronounced mitotic arrest compared to the control and 1 nM vinblastine (*Figure 6A*, Condition 6). Importantly, similar synergistic effects, characterized by elevated mitotic index and increased errors during metaphase and anaphase, were also observed upon MK2 depletion by siRNA in cells treated with 1 nM vinblastine, further reinforcing the notion that suppression of the MK2 signaling pathway can enhance MTA efficacy (*Figure 6—figure supplement 2A–C*). To investigate whether this synergy extends beyond mitosis, we next assessed the effects of MK2i and vinblastine co-treatment on TNBC cell migration. As expected, both MK2i and vinblastine individually impaired cell migration in CAL-51 cells; however, the combination treatment resulted in significantly greater inhibition compared to the individual treatment group or the control group (*Figure 6—figure supplement 3A-C*).

To better understand the impact of CMPD1 treatment on cancer cells at the gene expression level, we conducted RNA-seq analysis in MDA-MB-231 cells. When an FDR<0.05 was applied using DESeq2 for differential gene expression analysis, 351 genes were found to be upregulated, and 425 genes were downregulated 24 hr after treatment with 10 μM CMPD1 (*Figure 6B*, *Figure 6—figure supplement 4* and *Figure 6—source data 1*). Gene Ontology (GO) Biological Process (BP) pathway enrichment analysis revealed that the most significantly enriched pathways in upregulated genes relate to cell migration, while downregulated genes are predominantly associated with mitosis and chromosome segregation (*Figure 6C*, *Figure 6—figure supplement 4*). To explore whether unique pathways were up- or downregulated specifically in cancer cells, RNA-seq analysis was also performed on RPE1 cells. Comparing differentially expressed genes between MDA-MB-231 cells and RPE1 cells, cell death and apoptosis pathways were significantly enriched in genes uniquely upregulated in MDA-MB-231 cells (*Figure 6D* and *Figure 6—source data 2*). Genes specifically downregulated in MDA-MB-231 cells were again enriched in pathways related to mitosis and chromosome segregation (*Figure 6D*), consistent with the results that cancer cells are more sensitive to CMPD1 treatment (*Figure 2*). Collectively, CMPD1 upregulates cell death pathways while selectively downregulating mitotic genes in cancer cells, highlighting its potent cancer cell specificity. The pivotal role of the p38-MK2 signaling pathway in enhancing the efficacy of microtubule destabilizers likely contributes to these observed alterations in gene expression.

## Discussion

MTAs have been widely utilized as first- or second-line chemotherapy agents for various cancers; however, MTA-based treatment is often compromised by several limitations, including severe adverse effects and the development of drug resistance (*Brewer et al., 2016*; *Armstrong et al., 2006*; *Banerji et al., 2014*; *Lipton et al., 1989*; *Misawa et al., 2023*). Consequently, the development of novel therapeutic strategies to enhance the effectiveness of MTAs remains critical for improving clinical outcomes in cancer patients. In this study, we propose that inhibiting the p38-MK2 pathway can synergize with MTAs to significantly enhance their therapeutic efficacy (*Figure 7*). We demonstrate this synergistic effect by using a combination of an MK2-specific inhibitor with the microtubule depolymerizer vinblastine, alongside CMPD1, a dual-target inhibitor that simultaneously disrupts both the MK2 signaling pathway and microtubule dynamics (*Figure 6A*). Our findings reveal that CMPD1 exhibits a unique ability to depolymerize microtubules specifically at their plus ends, as well as selectively inducing mitotic defects and altering gene expression profiles in cancer cells (*Figures 2, 4D–H and 6B–D*). Furthermore, CMPD1 exhibited potency comparable to or greater than PTX in inhibiting

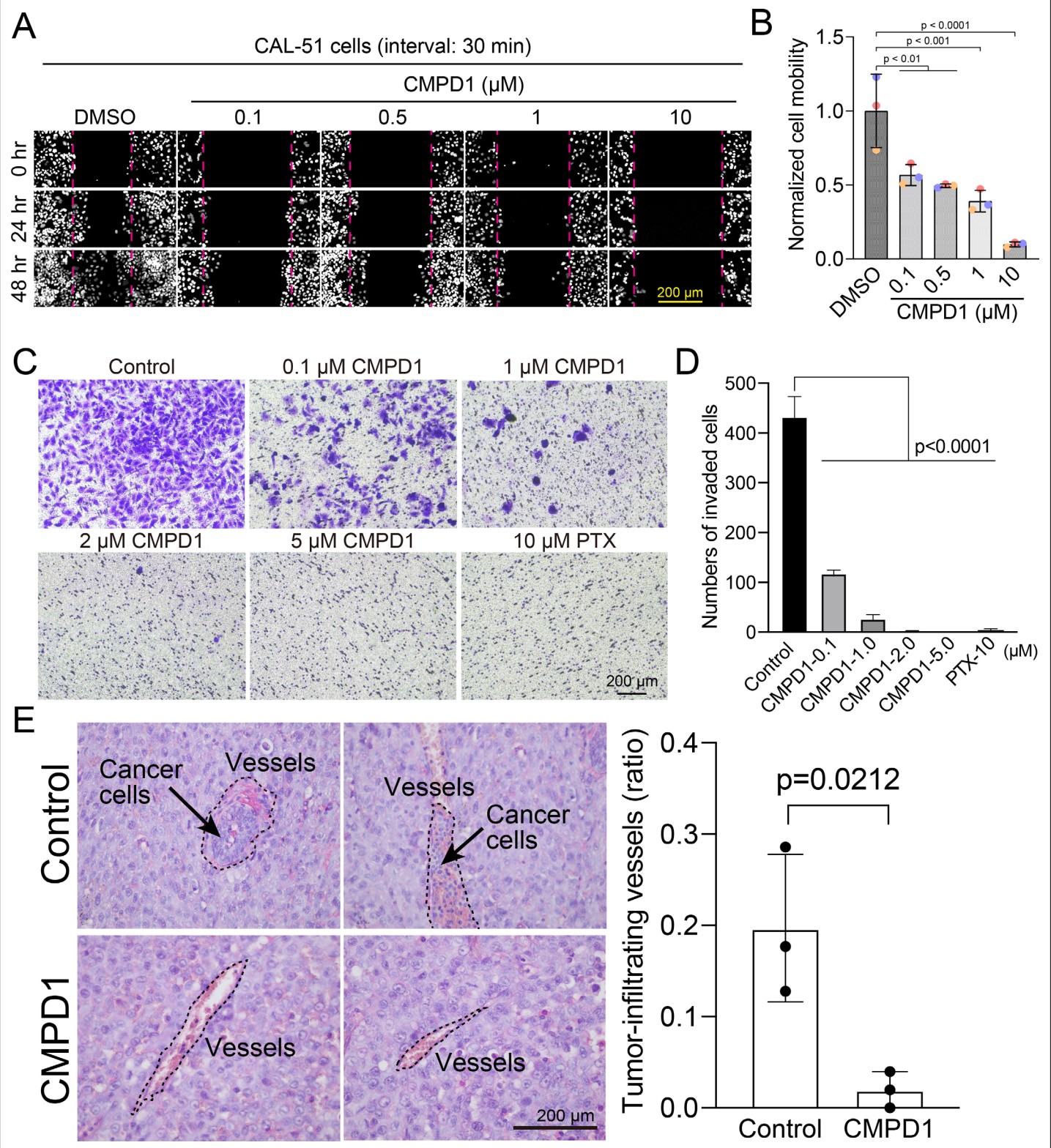

**Figure 5.** CMPD1 inhibits cancer cell migration and invasion. (**A**) Representative images of H2B-mScarlet CAL-51 cells treated with DMSO or CMPD1 at different concentrations (0, 0.1, 0.5, 1, 10 μM) at three different time points (0, 24, and 48 hr posttreatment). A wound (cell-free zone) was created using a tip, followed by the addition of indicated drugs and live-cell imaging (interval: 30 min). (**B**) Quantification of cell migration speed when cells were treated with indicated drugs. Each condition was normalized to the speed of DMSO-treated cells. The distance between the edge of the wound was measured using the ImageJ macro. Results are the mean ± s.d. N=3 biological replicates. The p-value was calculated using Tukey's multiple comparisons test.

*Figure 5 continued on next page*

*Figure 5 continued*

(**C**) Representative images of trans-well invasion assay using MDA-MB-231 cells treated with DMSO, CMPD1 (0.1, 1, 2, 5 µM), or 10 µM paclitaxel (PTX). The p-value was calculated using Tukey's multiple comparison test. (**D**) The quantification of the average number of invaded cells in each condition as shown in (**C**). (**E**) Left: example images of blood vessels in xenograft tumors derived from mice treated with DMSO or 1.0 mg/kg CMPD1 as shown in *Figure 3C*. Two sets of example images with ×400 magnification acquired from different tumors were shown. Right: the quantification of the percentage of blood vessels infiltrated with cancer cells. Results are mean ± s.d. The p-value was calculated using two-tailed t-test.

tumor growth both *in vitro* and *in vivo* (*Figure 3*). In terms of tumor specificity, PTX at clinically relevant concentrations lacks cancer selectivity (*Scribano et al., 2021*), whereas CMPD1, at similar low concentrations, specifically induces mitotic defects in breast cancer cells without affecting mitotic fidelity in non-transformed cells (*Figure 2*, *Figure 2—figure supplement 1A–C*). These findings suggest that p38-MK2 inhibition has the potential to enhance MTA efficacy, allowing for improved tumor growth inhibition at a lower MTA concentration, thereby reducing the risk of side effects.

To gain a deeper understanding of CMPD1's cancer cell selectivity, we conducted RNA-seq to compare global gene expression profiles between MDA-MB-231 and RPE1 cell lines, both with or without CMPD1 treatment (*Figure 6B*). CMPD1 treatment selectively upregulated gene pathways associated with cell death, aligning with its observed higher cytotoxicity to cancer cells. The cancer-specific selectivity of CMPD1 may also be attributed to differential expression levels of p38 and MK2. Previous studies have reported that MK2 is overexpressed in multiple myeloma (MM) and has the potential to serve as a marker of poor prognosis (*Guo et al., 2019*; *Gu et al., 2021*). Since MK2 facilitates the proliferation of MM cells by activating the Akt signaling pathway, depletion or inhibition of MK2 significantly impairs the growth of MM cells (*Guo et al., 2019*; *Gu et al., 2021*). Furthermore, p38, MK2, and phospho-MK2 are markedly upregulated in primary tumors of various cancers and exhibit an inverse correlation with overall survival rates (*Berggren et al., 2019*; *Kudaravalli et al., 2022*; *Morgan et al., 2022*; *Pomérance et al., 2006*; *Suresh et al., 2023*). Therefore, cancer cells with elevated MK2 or phospho-MK2 expression are likely more sensitive to treatments involving MK2 inhibitors. However, further studies are required to elucidate the detailed molecular mechanisms by which MK2 inhibitors enhance the efficacy of MTAs. Our study has demonstrated synergistic effects between MK2 inhibitors and MTAs, suggesting that the p38-MK2 pathway represents a promising therapeutic target in combination with MTAs for cancer chemotherapy.

# Materials and methods

**Key resources table**

| Reagent type (species) or resource | Designation | Source or reference | Identifiers | Additional information |
|---|---|---|---|---|
| Cell line (*Homo sapiens*) | hTERT RPE-1 | ATCC | CRL-4000, RRID:CVCL_4388CVCL_4388 | |
| Cell line (*Homo sapiens*) | MCF10A | ATCC | CRL-10317 | |
| Cell line (*Homo sapiens*) | CAL-51 | Scribano et al., Sci Transl Med. (2021) | RRID:CVCL_1110 | |
| Cell line (*Homo sapiens*) | Histone H2B-mScarlet, α-tubulin-mNeonGreen CAL-51 | Scribano et al., Sci Transl Med. (2021) | | |
| Cell line (*Homo sapiens*) | p53-KO CAL-51 | Redman-Rivera et al., Nat Commun. (2021) | | |
| Cell line (*Homo sapiens*) | MDA-MB-231 | ATCC | CRM-HTB-26 | |
| Cell line (*Homo sapiens*) | T47-D | ATCC | HTB-133 | |
| Transfected construct (human) | siRNA to MK2 (SMARTpool) | Horizon Discovery | L-003516-00-0010 | |

*Continued on next page*

*Continued*

| Reagent type (species) or resource | Designation | Source or reference | Identifiers | Additional information |
|---|---|---|---|---|
| Antibody | anti-Phospho-HSP27 (Ser82) (Rabbit monoclonal) | Cell Signaling Technology | Cat# 9709S, RRID:AB_11217429 | IF(1:1000) |
| Chemical compound, drug | CMPD1 | MedChemExpress | Cat# HY-108643 | |
| Chemical compound, drug | Paclitaxel | MedChemExpress | Cat# HY-B0015 | |
| Chemical compound, drug | Vinblastine | MedChemExpress | Cat# HY-13780 | |
| Chemical compound, drug | MK2-IN-3 hydrate (MK2i) | MedChemExpress | Cat# HY-112457 | |
| Chemical compound, drug | MG132 | MedChemExpress | Cat# HY-13259 | |
| Chemical compound, drug | MPS1 Inhibitor III, AZ3146 | Sigma | Cat# 5319760001 | |
| Chemical compound, drug | DMSO | Thermo Fisher Scientific | Cat# BP231-1 | |
| Commercial assay or kit | VitroGel Cell Invasion Assay Kit | TheWell Bioscienc | Cat# IA-VHM01-4P | |
| Other | Crystal Violet Stain Solution 1% Aqueous | Electron Microscopy Sciences | Cat# 26105-01 | |

## Cell culture

RPE1, MCF10A, CAL-51, MDA-MB-231, and T-47D cells were originally obtained from ATCC. Dulbecco's modified Eagle's medium (DMEM; Gibco) was used for culturing RPE1, CAL-51, MDA-MB-231, and T-47D cells, and DMEM/F12 supplemented with 5% (vol/vol) horse serum, 20 ng/ml hEGF, 0.5 mg/ml hydrocortisone, 100 ng/ml cholera toxin, 10 μg/ml insulin was used for culturing MCF10A cells. All the base media were additionally supplemented with 10% FBS (Sigma), 100 U/ml penicillin, and 100 mg/ml streptomycin (Gibco). All the cell lines we used were grown at 37°C in a humid incubator with 5% $CO_2$. Routine screening for *Mycoplasma* contamination was performed. CMPD1 (MCE, HY-108643), PTX (MCE, HY-B0015), vinblastine (MCE, HY-13780), MK2i (MCE, HY-112457), MG132 (MCE, HY-13259), and MPS1 inhibitor (Sigma, 5319760001) were dissolved in DMSO (Thermo, BP231-1). The concentration and the treatment duration of the above drugs are shown in figures or figure legends. For the washout assay, cells were treated with 2 μM CMPD1 for 4 hr, and then washed with pre-warmed complete media four times before starting live-cell imaging. For the siRNA-mediated knockdown experiment shown in *Figure 6—figure supplement 2*, Lipo-fectamin RNAiMAX Transfection Reagent (Invitrogen, 13778075) and ON-TARGETplus SMARTpool siRNA targeting MK2 (a mixture of four siRNAs; Horizon Discovery) were used according to the manufacturer's protocols.

## Imaging

For time-lapse image acquisition shown in *Figures 1C, 2A, F, and G*, *Figure 2—figure supplement 1A*, and *Figure 2—figure supplement 2A*, Z-stack images were obtained with a step size of 3 μm (12–15 μm in total) to cover entire mitotic cells using Nikon Elements software and a high-resolution Nikon Ti-2 inverted microscope equipped with a high-resolution Hamamatsu Flash V2 CMOS camera. The following objectives were used with the above scope: ×20 (NA 0.75, air) and ×40 (NA 1.25, silicon, for spindle depolymerization inside a cell). Images were taken every 2, 3, or 4 min for 24–72 hr. Cells were grown on glass bottom dishes with #1.5 coverslips and incubated in a Tokai Hit STX stage top incubator with PureBox Shiraito clean box (Tokai Hit). Feedback control of the Tokai Hit stage top incubator was used to stably maintain 37°C at growth medium. At least two biological replicates were performed. For fixed-cell experiments (immunofluorescence), cells were grown on high-precision #1.5 coverslips and imaged by a Nikon Ti-2 equipped with a Yokogawa SCU-W1-SoRa spinning disc confocal, Uniformizer, and a Hamamatsu Flash V3 CMOS camera. Quantitative image analysis was performed with Nikon Element (Nikon) and MetaMorph (Molecular Devices). No post-image processing was performed in all images. All imaging experiments were performed with biological replicates, repeated at least two to four times.

## Image analysis

Mitotic duration was defined as the time between NEBD and anaphase onset or mitotic exit if the cells underwent mitotic slippage. The DNA morphology visualized by histone H2B-EGFP or -mScarlet was used for the determination of mitotic substages. Those images were analyzed using Nikon Element. At least two to four biological replicates were performed.

## *In vitro* microtubule dynamics assays

Microtubules nucleated from GMPCPP-stabilized seeds were prepared and imaged as follows. A microtubule seed mixture was assembled from 50 μM tubulin (7:1:2, unlabeled:488-labeled: biotin-labeled) in BRB80 (80 mM PIPES, 1 mM $MgCl_2$, 1 mM EGTA, pH 6.9), clarified by centrifugation at 90 K rpm for 10 min, divided into 2 μl aliquots, snap-frozen in liquid nitrogen, and stored at –80°C. On the day of imaging, one of these aliquots was thawed, and to this, 0.4 μl 10 mM GMPCPP (Jena Biosciences; 1 mM final) and 1.6 μl BRB80 were added to bring the tubulin to a final concentration of 20 μM. The reaction was incubated at 37°C for 10 min (to promote polymerization) and stored at room temperature for up to 1 day. To prepare for an imaging session, 1 μl of the polymerized biotinylated-seed mixture was diluted 100× (to 200 nM final) into 79 μl BRB80 and 20 μl 0.75% methylcellulose (dissolved in BRB80), both at room temperature (the mixture was pipetted up and down to sheer seeds into smaller sizes). Flow chambers (~4–6 μl in volume) were assembled by adhering plasma-cleaned and biotin/PEGylated glass coverslips (as described in a previous study; *Chandradoss et al., 2014*) to glass slides using double-stick tape. Streptavidin (0.1 mg/ml; in BRB80) was introduced into the imaging chamber, after which the chamber was incubated at room temperature for 2–10 min. The chamber was washed with 10 μl BRB80 supplemented with 1% Pluronic and incubated for ~10–30 s, after which 10 μl of the diluted biotinylated-seed mixture was introduced. The imaging chamber was then immediately placed on a microscope stage pre-warmed to 37°C to monitor seed density in the chamber (we did our best to ensure each chamber had a similar density of seeds). Chambers were washed with 10 μl BRB80 supplemented with 1% Pluronic.

For the polymerization mix, a stock solution of 50 μM tubulin (12.5:1, unlabeled:rhodamine-labeled tubulin) in BRB80 was prepared, clarified by centrifugation at 90 K rpm for 10 min, divided into 20 μl aliquots, snap-frozen in liquid nitrogen, and stored at –80°C. Immediately prior to imaging, one of these aliquots was thawed and placed on ice. A 10 μl polymerization mix (final tubulin concentration, 15 μM) was assembled using the following: 3 μl 50 μM tubulin (from above), 0.5 μl 20 mM GTP, 0.5 μl 20 mM β-mercaptoethanol, 2 μl 0.75% methylcellulose, 0.5 μl glucose oxidase/catalase mix (1.2 mg glucose oxidase + 28.1 μl catalase + 99.1 μl BRB80)**,** 0.5 μl 30% glucose, 1 μl casein (from a 5% stock of alkali-soluble casein), and either 2 μl BRB80 or 2 μl of appropriate drug solution (20 μM CMPD1, or 5 μM vinblastine, each of which was prepared in BRB80). Note that all reagents were prepared or diluted in BRB80. The solution was gently mixed, warmed slightly between gloved fingertips for 10–20 s, and then immediately added to the flow chamber with the adhered GMPCPP-stabilized microtubule seeds (as described above). Chambers were then incubated for 15 min on the micro-scope stage at 37°C (to approach steady state), after which a freshly prepared polymerization mix with either tubulin only or tubulin and a drug was added to the chamber and imaged for 15–20 min. For experiments in which a drug was added mid-way through the imaging period (see kymographs in *Figure 4D*), we did the following: after the initial 15 min incubation with tubulin only, a fresh polym-erization mix without drug was added, and a 20 min movie was initiated. After 10 min, the movie was briefly paused, during which another freshly prepared polymerization mix with drug (or tubulin only) was added to the chamber, and the acquisition was then continued.

Microtubule dynamics were imaged using TIRFM on an inverted Nikon Ti-E using a 1.49 NA ×100 objective equipped with a Ti-S-E motorized stage (Nikon), piezo Z-control (Physik Instrumente), a motorized filter cube turret, a stage-top incubation system (LiveCell, Pathology Devices), and an iXon X3 DU897 cooled EM-CCD camera (Andor). We used 488 nm and 561 nm lasers (Nikon), a multi-pass quad filter cube set (C-TIRF for 405 nm/488 nm/561 nm/638 nm; Chroma), and emission filters mounted in a filter wheel (525 nm/50 nm, 600 nm/50 nm; Chroma). Images were acquired every 10 s. The microscope system was controlled by NIS-Elements software (Nikon), and images were analyzed using FIJI/ImageJ (NIH). Microtubule plus and minus ends were identified by exhibiting dynamics parameters distinct to each end (see kymographs in *Figure 4D* for examples).

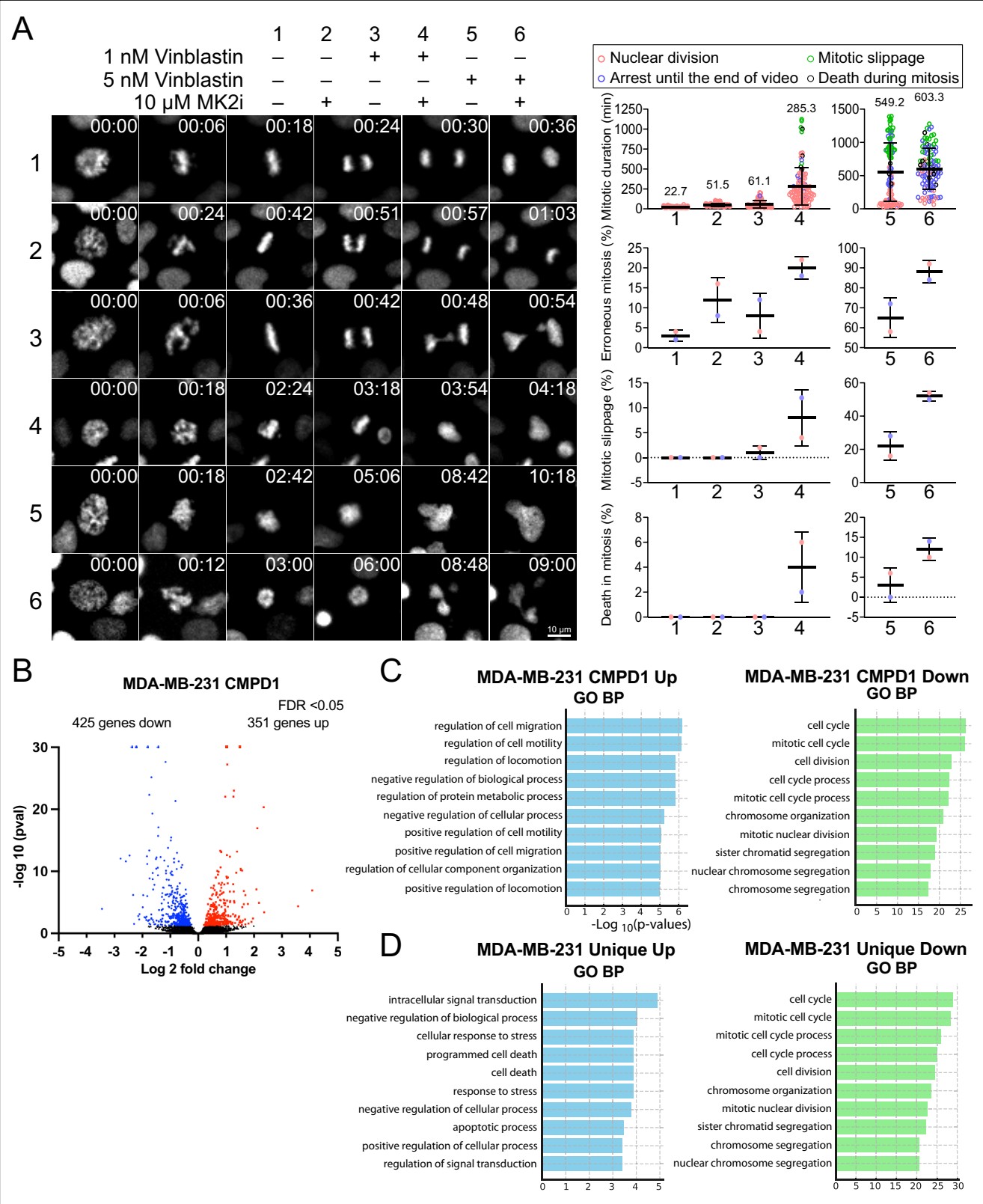

**Figure 6.** MK2 inhibition enhances the efficacy of microtubule inhibitors in cancer cells. (**A**) Left: representative time-lapse images of mitotic CAL-51 cells treated with indicated combinations of drugs (10 μM MK2i, 1 nM vinblastine, 5 nM vinblastine). Right: quantification of the mitotic duration, mitotic error rate, mitotic slippage rate, and the frequency of death in mitosis. The fate of mitotic cells was color-coded as indicated above the quantification plots. N=100 cells pooled from two biological replicates for each condition. Time shown on the upper right corner of the representative images is

*Figure 6 continued on next page*

*Figure 6 continued*

indicated in minutes after nuclear envelope breakdown (NEBD). (**B**) Volcano plot displaying changes in gene expression following CMPD1 treatment in MDA-MB-231 cells. RNA-seq was conducted with three biological replicates. Differentially expressed genes (DEGs) are highlighted in red (upregulated) and blue (downregulated). (**C**) Pathway enrichment analysis of DEGs using the Gene Ontology (GO) Biological Processes (BP). Enrichment analysis was performed with the DAVID online tool. (**D**) Comparison of GO BP enrichment analysis of DEGs unique to MDA-MB-231 cells relative to RPE1 cells. Genes uniquely up- or downregulated in MDA-MB-231 cells, but not in RPE1 cells, were subjected to GO BP enrichment analysis.

The online version of this article includes the following source data and figure supplement(s) for figure 6:

**Source data 1.** The list of differentially expressed genes in control MDA-MB-231 cells and in cells treated with CMPD1 or TPX.

**Source data 2.** The list of differential expressed genes upon treatment with CMPD1 or PTX in RPE1 cells.

**Figure supplement 1.** MK2 inhibitor suppresses the phosphorylation of downstream substrates without affecting the growth of cancer cells.

**Figure supplement 2.** MK2 depletion by siRNA recapitulates the effects of MK2i, which enhances vinblastine-induced cytotoxicity.

**Figure supplement 3.** MK2i and vinblastine synergistically suppress the migration ability of CAL-51 cells.

**Figure supplement 4.** Top 20 upregulated and downregulated genes in CMPD1-treated MDA-MB-231 cells.

## Anchorage-independent growth assay

Six-well plates (Eppendorf) were used for the anchorage-independent growth assay. 2 ml of cell culture media mixed with autoclaved 1% agarose was used to form a bottom layer in each well. After the bottom layer solidified, $2.4×10^4$ MDA-MB-231 cells were mixed with 2 ml of 0.8% agarose/media with or without CMPD1 or PTX and added to the top of the lower layer. After the top layer gel solidified, cells were incubated at 37°C in a humid atmosphere with 5% $CO_2$ for 3 weeks. 3 drops of the complete media were supplied in each well every 3 days to avoid drying out. Cells were stained by 100 µg/ml of iodonitrotetrazolium chloride solution (Sigma) prior to imaging.

## Cell viability assay

Cell viability was assessed using the MTT assay. Briefly, the same number of cells was seeded in a 24-well cell culture plate for each experiment. On day 2, the media in each well was replaced with those containing DMSO or different concentrations of CMPD1, and the incubation was continued for 2 days. On day 4, the media was replaced with the media containing 1 mg/ml MTT reagent (3-(4,5 dimethylthiazol-2-yl)-2,5diphenyltetrazolium bromide) and incubated for 3 hr. Viable cells have enzymes to reduce MTT to formazan, a purple product. Formazan was released from cells by adding

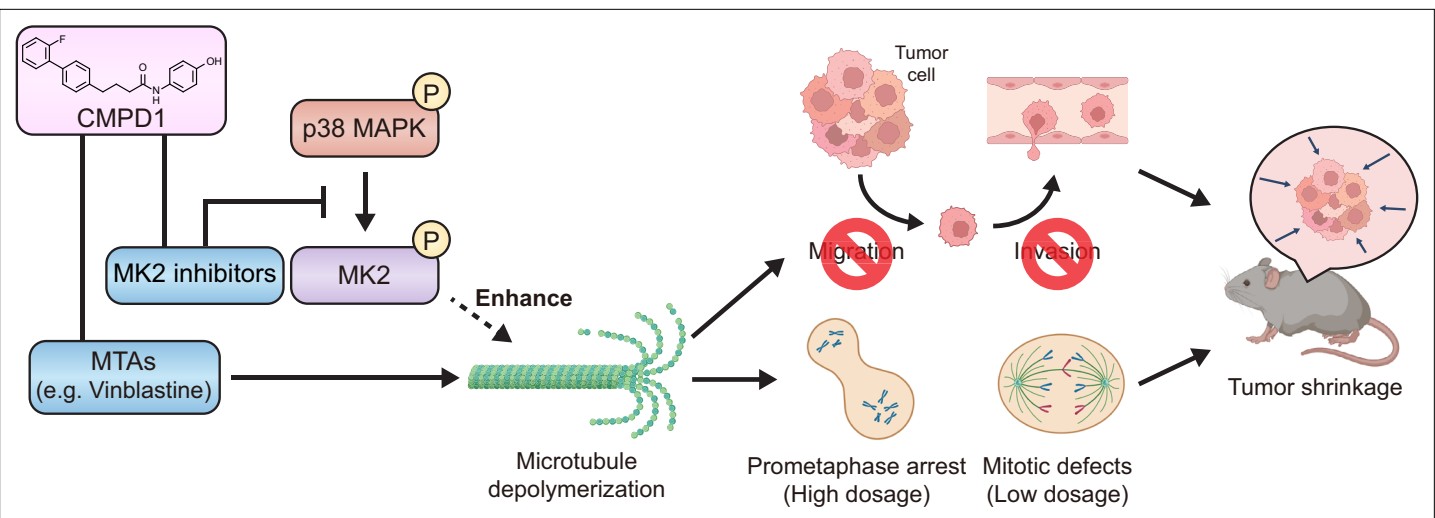

**Figure 7.** The model of CMPD1-mediated cytotoxicity effects on cancer cells. CMPD1 exhibits its tumor-specific cytotoxicity likely via two pathways. First, CMPD1 acts as a kinase inhibitor which prevents p38 mitogen-activated protein kinase (MAPK)-dependent phosphorylation of MK2, leading to the disruption of proper actin remodeling and spindle formation during mitosis. Second, CMPD1 serves as a microtubule-targeting agent (MTA), which can specifically induce depolymerization at the plus ends of microtubules. Both impaired actin reorganization and attenuated microtubule filament formation inhibit cancer cell migration and invasion, thereby preventing metastasis. On the other hand, failure of spindle assembly during mitosis causes the extended prometaphase arrest and the diminished mitotic fidelity, resulting in apoptosis of tumor cells and tumor shrinkage.

320 µl of DMSO and 40 µl Sorenson's glycine buffer (0.1 M glycine, 0.1 M NaCl [pH10.5]) to each well. The cell viability is assessed by measuring absorbance on a plate reader at 540 nm. $IC_{50}$ was calculated using GraphPad Prism (version 9.5).

## Cell growth assay

Same number of cells were seeded in 24-well cell culture plate. Cells were trypsinized and resuspended with cell culture media. Viable cells were stained by Trypan blue solution and counted under the microscope.

## Wound healing assay

Histone H2B-mscarlet CAL-51 cells were plated in an eight-well chamber slide (ibidi, 80807). 24 hr after cell seeding, a stripe of cell-free zone (the wound) was manually created on a highly confluent monolayer of cells using a sterile pipette tip. The cell culture media was then immediately replaced with a new one containing the indicated drug (DMSO, 0.1, 0.5, 1, and 10 µM CMPD1). Then, the cell locomotion dynamics was monitored using live-cell imaging at an interval of 30 min. To quantify the cell migration efficiency in each condition, we used an ImageJ macro to measure the distance between two boundaries of the wound. Briefly, the two boundaries of the wound were drawn and defined manually. Then, 40 lines perpendicular to the two boundaries were drawn automatically, and the length of each line was calculated and shown. The distance of cell movement was calculated by subtracting the distance at 20 hr post-drug treatment from the original distance at 0 hr. Each condition was normalized to the control (DMSO-treated cells).

## Cell invasion assay

CytoSelect 24-well cell invasion assay kit with basement membrane-coated inserts (Cell Biolabs, CBA-110) was used to assess the effects of CMPD1 and PTX on cell invasion. Following the overnight starvation of MDA-MB-231, the cells were seeded at $3.0 \times 10^4$ cells/well in the upper chamber and incubated in the media containing DMSO (control), CMPD1 (0.1, 1, 2, 5, 10 µM), or 10 µM PTX. The invasive cells passing through the basement membrane layer were stained, and the absorbance of each well was measured at 560 nm after extraction.

## Cell migration assay

VitroGel Cell Invasion Assay Kit (TheWell Bioscience, IA-VHM01-4P) is used to assess the synergistic effect of MK2i and Vinblastine on cell migration. We used VitroPrime Cell Culture Inserts (8 µm) to perform a transwell assay without using the hydrogel matrix provided by the kit. Following the overnight starvation of CAL-51 cells, the cells were seeded at $3.0 \times 10^4$ cells/well in the upper chamber and incubated in the media containing DMSO (control), 10 µM MK2i, 0.1 nM vinblastine, or 10 µM MK2i plus 0.1 nM vinblastine for 16 hr. The migrated cells passing through the membrane were fixed by PFA for 10 min and permeabilized by methanol for 2 min. Cells were stained by 1% crystal violet for 2 min and washed with PBS three times. After removing the cells at the upper side of the membrane using cotton swabs, cells located on the lower side of the membrane were imaged using a Nikon Ti Microscope equipped with Plan Fluor 10x DIC L N1 (NA = 0.30) and Nikon Digital Sight 10 camera.

## Tumor xenograft

CB17-Prkdc$^{scid}$/Jcl mice were used for the establishment of an orthotopic breast cancer model and therapy. This mice strain was purchased from CLEA Japan, Inc (Tokyo, Japan). Mice were maintained under specific pathogen-free conditions at Chiba University. All experimental procedures were performed in strict accordance with the National Institute of Health guidelines and were approved by the Institutional Animal Care and Use Committee of Chiba University. For *in vivo* experiments, sample sizes were determined on the basis of knowledge of inter-tumor growth rate variability, gained from previous model-specific experience. MDA-MB-231 and CAL-51 cells ($1 \times 10^6$ cells/0.1 ml) in 1:1 PBS:Matrigel were subcutaneously injected into the mammary fat pad of female mice. The experiment was performed twice. The first experiment used MDA-MB-231 cells, and the second experiment used both MDA-MB-231 and CAL-51 cells. In the first experiment, tumors were allowed to develop for 30 days, after which mice were randomly assigned to each treatment group, ensuring that baseline tumor volumes were balanced between treatment arms. The mice were treated with CMPD1 (i.p.,

15 µg/mouse/injection, 10 times for 3 weeks) or PTX (i.p., 5 mg/kg/injection, 10 times for 3 weeks). In the second experiment, treatment was started on the 10th day after transplantation, and the treatment drug was administered five times over 16 days until the 26th day. During the second experiment, weight was measured, blood was collected at the end of the study, and the liver and kidneys were also removed for further analysis. Both maximum and minimum diameters of the resulting tumors were measured every other day using a slide caliper. Tumor volumes were calculated as previously described. Mice were euthanized via $CO_2$ inhalation. The maximum tumor diameter permitted under the relevant animal protocols is 25 mm, and this limit was not exceeded in any experiment.

## H&E staining

Hematoxylin and eosin (H&E) staining was performed using tissue samples obtained from the xenograft model. Tissue samples were thin sliced to a thickness of 4 µm. For staining, slides were first stained with hematoxylin for 5 min, followed by eosin staining for 2 min. The slides were then dehydrated through a series of ethanol solutions, cleared in xylene, and mounted. The staining was evaluated based on the tissue morphology.

## Blood invasion evaluation

All slides from mouse tumors with or without CMPD1 treatment were stained with H&E stain and were screened for vascular invasion using strict criteria based on previous reports (*Pinder et al., 1994*; *Lauria et al., 1995*). All slides were blindly evaluated by two investigators (MT and HY). All these vascular invasions were adopted only if they were picked up on H&E staining. Vascular invasion was determined by observing tumor cells within blood vessels. The cases were categorized as blood vessel invasion positive or negative. Representative histological images of positive and negative blood vessel invasion status are shown in *Figure 5E*.

## Biochemical analysis of animal models

Whole blood samples were collected from all mice, and blood biochemistry determinations were performed with an Automatic Analyzer Model 7070 (Hitachi Co., Ltd., Tokyo, Japan). Parameters were aspartate aminotransferase, alanine aminotransferase, γ-glutamyl transpeptidase, blood urea nitrogen, and creatinine (Oriental Yeast Co., Ltd., Tokyo, Japan).

## Giemsa staining

Slides were fixed in absolute methanol for 30 s to facilitate cell attachment and preserve optimal staining characteristics, then allowed to air-dry. The slides were immersed in Wright-Giemsa stain (container 1) for 60 s without agitation. For the Rapid Wright's One-Step Stain reagent, the staining duration was reduced to 15–30 s. Excess stain was removed by draining or blotting the edges of the slides, avoiding direct contact with the smear. The slides were then immersed in buffer solution (container 2) for 60 s (15–45 s for the RapidWright's reagent), followed by draining to remove excess buffer. Subsequently, the slides were dipped in rinse solution (container 3) for 2–10 s, with the RapidWright's reagent requiring quick dips for 25 s. Excessive buffering and rinsing were avoided to prevent decolorization. The slides were air-dried in a vertical position on a paper towel. For microscopic analysis, leukocytes were counted per field at low magnification, ensuring that fields were selected uniformly across specimens.

## FACS

MDA-MB-231 cells were incubated with or without CMPD1 for 24 hr. Cells were then twice washed with PBS. Approximately $2 \times 10^6$ cells were fixed with ice-cold ethanol for 16 hr. Samples were washed with cold PBS and stained with PI (Sigma) containing Triton-X (Sigma) and DNAse-free RNAse A (Sigma) in PBS for 30 min. Then, samples were measured by a flow cytometer.

## RNA-seq

24 hr after 10 µM CMPD1 treatment, RNAs were purified by the RNeasy Mini Kit (QIAGEN). For MDA-MB-231 cells, RNA-seq library preparation was conducted with the QuantSeq 3' mRNA-Seq Library Prep Kit FWD for Illumina (LEXOGEN), and the libraries were sequenced on a NextSeq500 at Kazusa DNA Research Institute. In the case of RPE1 cells, sequencing libraries were prepared using

the TruSeq Stranded mRNA Kit and sequenced on a NovaSeq 6000 at Macrogen Japan. Adapter sequences were removed from the raw sequencing data, and after adapter trimming, reads were mapped to the human reference genome (GRCh38) using the STAR aligner (*Dobin et al., 2013*). Read counts for each gene were collected by featureCounts (version v1.6.4) (*Liao et al., 2014*). Differentially expressed genes were identified with DESeq2 (*Love et al., 2014*) using filtering thresholds of FDR<0.05. Pathway enrichment analysis was performed by DAVID (*Huang et al., 2009*).

## Statistics

Statistical significance was determined using two-tailed unpaired Student's t-test for comparison between two independent groups or one-way ANOVA for multiple comparisons. For TIRF experiments in *Figure 2G–I*, p-values were calculated from Z scores (*Figure 2G*; as previously described in *Marzo et al., 2019*), Mann-Whitney tests (*Figure 2H and I*, left and right), or by unpaired two-tailed Welch's t-tests (*Figure 2H and I*, middle). The latter two tests were selected as follows: the unpaired two-tailed Welch's t-test was used when the datasets were determined to be normal (by the D'Agostino and Pearson test for normality; p>0.05). In the case where only one (or neither) was determined to be normal (p<0.05), the Mann-Whitney test was used. For significance, *p<0.05, **p<0.01, ***p<0.001, ****p<0.0001 were considered statistically significant. All quantifications were based on at least two independent biological replicates.

## Acknowledgements

We thank Drs. Most S Parvin, Ainslie Homan, Ozge Ali, Will Rossman, Masaharu Kasuya, Wang Junchao, Randy Owen, and Alex Li for help with data and image analysis. We also thank Drs. Terry Juang, Albert Wang, and Caleb Carlsen for providing technical guidance and experiment materials. CAL-51 and CAL-51 p53 KO cells were kindly gifted from Drs. Beth Weaver, Mark Burkard, and Jennifer A Pietenpol. Part of this work was supported by Wisconsin Partnership Program, Research Forward from the Office of the Vice Chancellor for Research (OVCR) and Wisconsin Alumni Research Foundation (WARF), start-up funding from University of Wisconsin-Madison SMPH, UW Carbone Cancer Center, and McArdle Laboratory for Cancer Research, and NIH grant R35 GM147525 and U54 AI170660 (to AS), NIH P20GM104360 and UND School of Medicine & Health Sciences Dean's fund (to M Takaku), JSPS KAKENHI 21K08638 (to M Takada), R35 GM130365 and NSF1518083 (to JG DeLuca), and R35 GM139483 and NSF1518083 (to S Markus).

## Additional information

### Funding

| Funder | Grant reference number | Author |
|---|---|---|
| National Institutes of Health | R35 GM147525 | Aussie Suzuki |
| National Institutes of Health | U54 AI170660 | Aussie Suzuki |
| National Institutes of Health | R35 GM130365 | Jennifer G DeLuca |
| University of Wisconsin-Madison | Research Forward | Aussie Suzuki |
| National Institutes of Health | P20GM104360 | Motoki Takaku |
| National Institutes of Health | R35 GM139483 | Steven M Markus |
| National Science Foundation | NSF1518083 | Jennifer G DeLuca Steven M Markus |
| Japan Society for the Promotion of Science | KAKENHI 21K08638 | Mamoru Takada |

| Funder | Grant reference number | Author |
|---|---|---|
| University of North Dakota | UND School of Medicine & Health Sciences Dean's fund | Motoki Takaku |

The funders had no role in study design, data collection and interpretation, or the decision to submit the work for publication.

### Author contributions
Yu-Chia Chen, Data curation, Formal analysis, Investigation, Visualization, Methodology, Writing – original draft; Mamoru Takada, Data curation, Formal analysis, Funding acquisition, Investigation, Visualization, Methodology, Writing – review and editing, Writing – original draft; Aerica Nagornyuk, Muhan Yu, Hideyuki Yamada, Takeshi Nagashima, Masayuki Ohtsuka, Data curation, Formal analysis, Investigation, Writing – review and editing; Jennifer G DeLuca, Steven M Markus, Motoki Takaku, Data curation, Formal analysis, Funding acquisition, Validation, Investigation, Visualization, Methodology, Writing – review and editing; Aussie Suzuki, Conceptualization, Data curation, Supervision, Funding acquisition, Visualization, Writing – original draft, Project administration

### Author ORCIDs
Yu-Chia Chen https://orcid.org/0000-0002-2404-7597
Mamoru Takada https://orcid.org/0000-0002-3961-9009
Jennifer G DeLuca https://orcid.org/0000-0002-3598-1721
Steven M Markus https://orcid.org/0000-0002-3098-0236
Motoki Takaku https://orcid.org/0000-0002-8652-2541
Aussie Suzuki https://orcid.org/0000-0001-7390-5116

### Ethics
All animal experiments were reviewed and ethically approved by the Institutional Animal Care and Use Committee in Chiba University, Japan (Dou30-423), and were carried out in accordance with the regulations in the Guide for the Care and Use of Animals in Research of Japan.

Reviewer #1 (Public review): https://doi.org/10.7554/eLife.104859.3.sa1
Reviewer #2 (Public review): https://doi.org/10.7554/eLife.104859.3.sa2
Reviewer #3 (Public review): https://doi.org/10.7554/eLife.104859.3.sa3
Author response https://doi.org/10.7554/eLife.104859.3.sa4

## Additional files

### Supplementary files
MDAR checklist

### Data availability
RNA-seq data generated in this study are available at Gene Expression Omnibus under Accession Number: GSE224462: Additional source data are available at Dryad (https://doi.org/10.5061/dryad.m37pvmdf1).

The following dataset was generated:

| Author(s) | Year | Dataset title | Dataset URL | Database and Identifier |
|---|---|---|---|---|
| Suzuki A | 2025 | Inhibition of p38-MK2 pathway enhances the efficacy of microtubule inhibitors in breast cancer cells | https://doi.org/10.5061/dryad.m37pvmdf1 | Dryad Digital Repository, 10.5061/dryad.m37pvmdf1 |

The following previously published dataset was used:

| Author(s) | Year | Dataset title | Dataset URL | Database and Identifier |
|---|---|---|---|---|
| Takaku M, Takada M, Suzuki A | 2024 | CMPD1 is a microtubule inhibitor with tumor-specific cytotoxicity | https://www.ncbi.nlm.nih.gov/geo/query/acc.cgi?acc=GSE224462 | NCBI Gene Expression Omnibus, GSE224462 |

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
