## [Editor Report · eLife Assessment]

This study provides **valuable** findings that MK2 inhibitor CMPD1 can inhibit the growth, migration and invasion of breast cancer cells both in vitro and in vivo. The evidence supporting the claims of the authors is **solid**, although the detailed molecular mechanism and additional animal experiments would strengthen the paper. This study will be of interest to the breast cancer field.

---

## [Referee Report · Reviewer #1 (Public review)]

In this paper, the authors reveal that the MK2 inhibitor CMPD1 can inhibit the growth, migration and invasion of breast cancer cells both in vitro and in vivo by inducing microtubule depolymerization, preferentially at the microtubule plus-end, leading to cell division arrest, mitotic defects, and apoptotic cell death. They also showed that CMPD1 treatment upregulates genes associated with cell migration and cell death, and downregulates genes related to mitosis and chromosome segregation in breast cancer cells, suggesting a potential mechanism of CMPD1 inhibition in breast cancer. Besides, they used the combination of an MK2-specific inhibitor, MK2-IN-3, with the microtubule depolymerizer vinblastine to simultaneously disrupt both the MK2 signaling pathway and microtubule dynamics, and they claim that inhibiting the p38-MK2 pathway may help to enhance the efficacy of MTAs in the treatment of breast cancer.

---

## [Referee Report · Reviewer #2 (Public review)]

Summary:

This study explores the potential of inhibiting the p38-MK2 signaling pathway to enhance the efficacy of microtubule-targeting agents (MTAs) in breast cancer treatment using a dual-target inhibitor.

Strengths:

The study identifies the p38-MK2 pathway as a promising target to enhance the efficacy of microtubule-targeting agents (MTAs), offering a novel therapeutic strategy for breast cancer treatment. The study also employs a wide range of techniques, especially live-cell imaging, to assess the microtubule dynamics in TNBC cells. The revised manuscript added new in vitro and in vivo evidence that furtherly supported the conclusions.

Comments on revisions:

The authors have appropriately addressed all of my comments and concerns. Specifically, they performed additional in vitro experiments using MCF10A cells and p53 knockout cells to determine the IC50 of CMPD1. They also repeated the in vivo treatment experiment and evaluated the toxicity of the drug treatment in the CAL-51 model. Furthermore, they provided genetic evidence for the combination treatment. I'm satisfied with the revision and have no further major comments. Minor comment: make sure the name of the chemo drug shown in Fig. 3 is consistent.

---

## [Referee Report · Reviewer #3 (Public review)]

Summary:

The authors demonstrated MK2i could enhance the therapeutic efficacy of MTAs. With the tumour xenograft and migration assay, the author suggested that the p38-MK2 pathway may serve as a promising therapeutic target in combination with MTAs in cancer treatment.

Strengths:

The authors provided a potential treatment for breast cancer.

Comments on revisions:

A xenograft experiment should be included to evaluate the synergistic effect of MK2i and vinblastine.

---

## [Author Response]

The following is the authors’ response to the original reviews

**Public Reviews:**

**Reviewer #1 (Public review):**
In this paper, the authors reveal that the MK2 inhibitor CMPD1 can inhibit the growth, migration, and invasion of breast cancer cells both in vitro and in vivo by inducing microtubule depolymerization, preferentially at the microtubule plus-end, leading to cell division arrest, mitotic defects, and apoptotic cell death. They also showed that CMPD1 treatment upregulates genes associated with cell migration and cell death, and downregulates genes related to mitosis and chromosome segregation in breast cancer cells, suggesting a potential mechanism of CMPD1 inhibition in breast cancer. Besides, they used the combination of an MK2-specific inhibitor, MK2-IN-3, with the microtubule depolymerizer vinblastine to simultaneously disrupt both the MK2 signaling pathway and microtubule dynamics, and they claim that inhibiting the p38-MK2 pathway may help to enhance the efficacy of MTAs in the treatment of breast cancer. However, there are a few concerns, including:(1) What is the effect of CMPD1 on breast cancer metastasis?

In this study, we hypothesized that the MK2 signaling pathway could synergize with microtubule-targeting agents (MTAs) to enhance anti-cancer efficacy. We utilized CMPD1 as a potent dual-function inhibitor, targeting both MK2 and microtubule dynamics. By simultaneously inhibiting these pathways, CMPD1 not only shows the therapeutic impact of MTAs, but also significantly suppresses breast cancer cell migration and invasion. Therefore, we propose that CMPD1, through its dual inhibition of MK2 activity and microtubule dynamics, may offer enhanced specificity and efficacy in preventing breast cancer metastasis and limiting tumor progression.

(2) The mechanism is lacking as to how MK2 inhibitors enhance the efficacy of MTAs.

Thank you for the valuable suggestion. We agree that our current findings do not fully elucidate the underlying mechanism by which MK2 inhibition synergistically enhances the efficacy of MTAs. We recognize this as an important area for further investigation and are committed to exploring the molecular interplay between MK2 signaling and microtubule dynamics in future studies. A deeper mechanistic understanding will be critical to establishing a strong rationale for the potential co-treatment of MK2 inhibitors and MTAs in clinical breast cancer therapy.

**Reviewer #2 (Public review):**
Summary:This study explores the potential of inhibiting the p38-MK2 signaling pathway to enhance the efficacy of microtubule-targeting agents (MTAs) in breast cancer treatment using a dual-target inhibitor.Strengths:The study identifies the p38-MK2 pathway as a promising target to enhance the efficacy of microtubule-targeting agents (MTAs), offering a novel therapeutic strategy for breast cancer treatment. In addition, the study employs a wide range of techniques, especially live-cell imaging, to assess the microtubule dynamics in TNBC cells.

We sincerely appreciate your recognition of the significance and impact of our work.

Weaknesses:The study primarily uses RPE1 cells as the control for normal cells, which may not fully capture the response of normal mammary epithelial cells. While CMPD1 is shown to be effective in suppressing tumor growth in MDA-MB-231 xenograft, the study lacks detailed toxicity data to confirm its safety profile in vivo.

Thank you for your valuable suggestions. In the revised manuscript, we have included CMPD1 treatment in MCF10A cells, a more appropriate non-transformed control line commonly used in breast cancer research. Notably, MCF10A cells exhibited results similar to those observed in RPE1 cells, further reinforcing our conclusion that breast cancer cells display increased sensitivity to CMPD1 treatment. These new findings are presented in Figure 2-Supplement 1A-C. Additionally, we performed further xenograft experiments using CAL-51 and MDA-MB-231 cells. We collected data on tumor growth, mouse body weight, survival rates, and other relevant parameters to comprehensively assess toxicity. The newly obtained results are presented in Figure 3F-G and Figure 3-Supplement 1-3.

**Reviewer #3 (Public review):**
Summary:The authors demonstrated MK2i could enhance the therapeutic efficacy of MTAs. With Tumor xenograft and migration assay, the author suggested that the p38-MK2 pathway may serve as a promising therapeutic target in combination with MTAs in cancer treatment.Strengths:The authors provided a potential treatment for breast cancer.

Thank you for recognizing the importance and significance of our work.

Weaknesses:(1) In Figure 2, the authors used a human retinal pigment epithelial-1 (RPE1) cell line to show that breast cancer cells are more sensitive to CMPD1 treatment. MCF10A cells would be suggested here as a suitable control. Besides, to compare the sensitivity, IC50 indifferent cell lines should be measured.

In the revised manuscript, we have addressed these points by determining the IC50 values for CMPD1 in MDA-MB-231, CAL-51, MCF10A, and CAL-51 p53 knockout cells. These new results are presented in Figure 2-Supplement Figure 3.

(2) The data of MDA-MB-231 in Figure 1D is not consistent with CAL-51 and T47D, also not consistent with the data in Figures 2B-C.

In the revised manuscript, we have included all relevant statistical analyses in Figure 1D. In MDA-MB-231 cells, there are no statistically significant differences in mitotic duration between 1 µM and 5 µM, 5 µM and 10 µM, or 1 µM and 10 µM CMPD1 treatments. Similarly, no significant differences are observed between 1 µM and 5 µM or 5 µM and 10 µM CMPD1 treatments in CAL-51 cells, and between 5 µM and 10 µM in T-47D cells. These results suggest that mitotic duration does not exhibit a clear dose-dependent relationship within the 1–10 µM range, likely because mitotic arrest has reached a near-plateau effect at these concentrations.

It is also important to note that the experimental conditions in Figures 1 and 2 are fundamentally different. Figure 1 investigates the effects of higher concentrations of CMPD1 (≥1 µM), which severely disrupt microtubule organization and result in robust mitotic arrest, with cells arrested in mitosis for over 8 hours. In contrast, the conditions in Figure 2 utilize much lower concentrations of CMPD1 (10–50 nM), which are insufficient to cause complete microtubule depolymerization, but are capable of inducing a subtle yet statistically significant mitotic delay, particularly in breast cancer cell lines. These lower concentrations were chosen to mimic clinically relevant intratumoral drug levels. Previous studies have reported that paclitaxel (PTX) concentrations in patient tumors approximate ~50 nM when modeled in vitro. At these physiologically relevant levels, PTX does not induce strong mitotic arrest but instead causes moderate delays that result in division errors and chromosomal instability, ultimately contributing to cancer cell death. In this study, the conditions used in Figure 2 emulate these clinically relevant concentrations for CMPD1. We found that, similar to PTX, low-dose CMPD1 induces a slight but significant mitotic delay without triggering a full mitotic arrest. Notably, unlike PTX, CMPD1 appears to exert this effect selectively in breast cancer cells, contributing to mitotic errors and potentially enhancing therapeutic efficacy through targeted chromosomal instability.

(3) To support the authors' conclusion in Figure 5, an additional animal experiment performed by tail vein injection would be helpful.

While current technical limitations have precluded us from conducting this suggested experiment in this study, we have performed complementary xenograft studies using CAL-51 cells treated with CMPD1. These experiments included a comprehensive toxicity analysis. Furthermore, we carried out an in vitro migration assay using CAL-51 cells under combined treatment with the MK2 inhibitor and vinblastine. These additional findings are presented in Figure 3–Supplement 1–3 and Figure 6–Supplement 3. We recognize the importance of the suggested tail vein injection approach and are actively pursuing further mechanistic studies, including this experiment, in our ongoing and future work.

(4) Page 14, to evaluate the combination result of MK2i and vinblastine, an in vivo animal assay must be performed.

We appreciate the reviewer’s valuable suggestion. We are actively investigating the synergistic mechanisms between the MK2 inhibitor and microtubule-targeting agents (MTAs). In future studies, we plan to extend our findings by conducting xenograft experiments to further evaluate their therapeutic potential in vivo.

(5) The authors used RNA-seq to show some pathways affected by CMPD1. What are the key/top genes that were affected? How about the mechanism?

In the revised manuscript, we have included the top 20 upregulated and downregulated genes identified from RNA-seq analysis using MDA-MB-231 cells. This new data is presented in Figure 6-Supplement Figure 4. Gene Ontology (GO) Biological Process (BP) pathway enrichment analysis revealed that the most significantly enriched pathways among upregulated genes are associated with cell migration, whereas the downregulated genes are primarily involved in mitosis and chromosome segregation. These transcriptional changes are consistent with the phenotypic outcomes observed in our experiments, supporting the functional relevance of CMPD1 treatment. However, further investigation will be necessary to elucidate the detailed molecular mechanisms underlying these effects.

(6) Line 127, more experiments should be involved to support the conclusion.

In the revised manuscript, we have addressed this point by performing additional experiments, including determination of the IC₅₀ values of CMPD1 in MDA-MB-231, CAL-51, MCF10A, and CAL-51 p53 knockout cells. We also conducted live-cell imaging analyses using MCF10A cells. These new results further reinforce our conclusion that breast cancer cells are more sensitive to CMPD1 treatment than normal breast epithelial cells, and that this sensitivity is independent of p53 status. The new data are presented in Figure 2-Supplement Figures 1 and 3.

**Recommendations for the authors:**

**Reviewer #1 (Recommendations for the authors):**
(1) Figure 1D: As the concentration of CMPD1 increased, the mitotic duration of MDA-MB-231 cells decreased, why was that?

Although there appears to be a slight decrease in mitotic duration with increasing concentrations of CMPD1, our quantitative analysis reveals no statistically significant differences among the 1 to 10 µM treatment groups in MDA-MB-231 cells. In the revised manuscript, we have included all relevant statistical analyses in Figure 1D for clarity. Importantly, all CMPD1-treated groups exhibit a pronounced and statistically significant prolongation of mitosis compared to the DMSO-treated control. While the average mitotic duration in control cells is approximately 30 minutes, cells exposed to 1–10 µM CMPD1 consistently display mitotic durations exceeding 8 hours, indicating a strong and sustained mitotic arrest across this concentration range.

**Reviewer #2 (Recommendations for the authors):**
(1) The rationale for using RPE1 as normal cell control instead of normal mammary epithelial cells as control is unclear. Using normal mammary epithelial cells such as MCF10A for the study is recommended.

Thank you for this valuable suggestion. In the revised manuscript, we have included additional experiments using non-transformed mammary epithelial MCF10A cells. The new data, presented in Figure 2-Supplement Figures 1 and 3, include both IC50 measurements and live-cell imaging analyses. These results further support our conclusion that breast cancer cells are significantly more sensitive to CMPD1 treatment compared to normal mammary epithelial cells.

(2) It is intriguing that CAL-51 cells are more sensitive to CMPD1 than MDA-MB-231 cells; examining how p53 signaling changes in these cells would be worthwhile.

We appreciate this insightful comment. In the revised manuscript, we have measured the IC₅₀ values for both CAL-51 and CAL-51 p53 knockout (p53KO) cells. The results show no significant difference in CMPD1 sensitivity between the two, suggesting that the enhanced sensitivity of CAL-51 cells is independent of p53 status. These new findings are presented in Figure 2—Supplement Figure 3.

(3) Figures S1A and B are not described and cited in the main text.

We apologize for this oversight. In the revised manuscript, we have correctly cited and described Figures S1A and B (Figure 2-Supplement Figure 2 A-B in revised manuscript) in the main text.

(4) I'm not that convinced by the conclusion made from Lines 201-204. First, Figure S2C, which is the growth of tumor volume, does not reflect the toxicity of the drug treatment. No additional data evaluating the toxicity (such as body weight change) under the regimen was shown. Second, although the tumor weight by the endpoint indicated some anti-tumor effect in the MDA-MB-231 xenograft model, the tumor volume does not show the same pattern (the dot lines do not well distinguish which group from which). I would suggest repeating the in vivo experiment using CAL-51 cells since it is more sensitive to CMPD1 according to the previous data.

Thank you for this thoughtful and constructive feedback. In the revised manuscript, we have addressed these concerns through several additional experiments. We performed new xenograft studies using CAL-51 TNBC cells, in parallel with further toxicity-focused analyses in the MDA-MB-231 model. Consistent with previous results, CMPD1 treatment significantly suppressed tumor growth in CAL-51 xenografts (Figure 3F-G), further supporting its efficacy in a more sensitive cell line. To evaluate drug-associated toxicity, we measured body weight changes throughout the course of treatment. CMPD1-treated mice maintained a comparable weight gain to the control group, whereas mice treated with paclitaxel (PTX) showed significantly reduced body weight (Figure 3-Supplement Figure 2A). Notably, animal deaths occurred only in the PTX-treated groups in both MDA-MB-231 and CAL-51 models (Figure 3-Supplement Figure 2B). We also assessed organ toxicity, including both anatomical and functional evaluations of the kidney and liver, and observed no significant damage in CMPD1-treated mice (Figure 3-Supplement Figures 3A-B and 3D). Furthermore, white blood cell (WBC) counts remained stable in the CMPD1 group, while PTX treatment led to a significant reduction (Figure 3-Supplement Figures 3C-D). These additional data provide strong evidence for the anti-tumor efficacy and lower toxicity of CMPD1 in vivo.

(5) While I appreciate the combination effect of treating cells with the MK2 inhibitor with vinblastine. I would consider using genetic knockdown as a complementary approach to demonstrate that inhibiting the p38-MK2 pathway synergized with microtubule depolymerizing agents. In addition, could inhibition of the p38-MK2 pathway alone induce the cell growth inhibition observed with CMPD1 treatment?

Thank you for these important suggestions. In the revised manuscript, we have incorporated siRNA-mediated knockdown of MK2 in combination with vinblastine treatment. This genetic approach revealed synergistic effects on mitotic index and mitotic errors, closely mirroring the phenotypes observed with pharmacological co-treatment using the MK2 inhibitor and vinblastine (Figure 6-Supplement Figure 2A-C). These results further validate the role of the p38-MK2 pathway in modulating mitotic progression in the presence of MTAs. To address whether MK2 inhibition alone is sufficient to impair cell growth, we performed validation experiments using the MK2 inhibitor at 10 µM. At this concentration, the inhibitor effectively blocked phosphorylation of Hsp27, a major downstream substrate of MK2, under H2O2-induced ROS stress conditions (Figure 6-Supplement Figure 1A-B), confirming MK2 signaling pathway inhibition. However, treatment with the MK2 inhibitor alone did not significantly affect cell proliferation, as shown by a 4-day growth curve analysis in CAL-51 cells (Figure 6-Supplement Figure 1C). These findings suggest that inhibition of the p38-MK2 pathway alone is not sufficient to suppress cancer cell growth, and that its synergistic interaction with MTAs, such as vinblastine, is essential for the observed anti-proliferative effects.

(6) Phenotypic studies (such as anchorage-independent growth and cell migration and invasion assay) of combining MK2 inhibitor with vinblastine in TNBC cells are recommended.

Thank you for this valuable suggestion. In the revised manuscript, we have conducted cancer cell migration assays using CAL-51 TNBC cells treated with control, MK2 inhibitor alone, vinblastine alone, or the combination of both. Our results demonstrate that the combination treatment significantly enhances the inhibition of cell migration compared to either agent alone (Figure 6-Supplement Figure 3A-C). These findings provide additional phenotypic evidence supporting the synergistic interaction between MK2 inhibition and microtubule-targeting agents in TNBC cells.

**Reviewer #3 (Recommendations for the authors):**
The authors can utilize diverse experiments to support their conclusions.

Thank you for this important suggestion. In the revised manuscript, we have conducted a series of additional experiments to robustly support our conclusions.

These include:

(1) Xenograft studies using CAL-51 TNBC cells, along with comprehensive toxicity evaluations.

(2) CMPD1 sensitivity analysis in non-transformed MCF10A mammary epithelial cells.

(3) IC50 measurements in MDA-MB-231, CAL-51, CAL-51 p53 knockout, and MCF10A cells.

(4) Cell migration assays assessing the combination effects of MK2 inhibitor and vinblastine

(5) siRNA-mediated genetic knockdown of MK2 to complement pharmacological findings

Collectively, these additional data sets substantially strengthen the evidence base for our conclusions and provide a more comprehensive mechanistic understanding.